



# Shallow groundwater level time series and a groundwater chemistry survey from a boreal headwater catchment

Jana Erdbrügger[1], Ilja van Meerveld[1], Jan Seibert[1, 2], Kevin Bishop[2]

[1]Department of Geography, University of Zurich, Zurich, Switzerland
[2]Department of Aquatic Sciences and Assessment, Swedish University of Agricultural Sciences, Uppsala, Sweden

*Correspondence to*: Jana Erdbrügger (jana.erdbruegger@geo.uzh.ch)

**Abstract.** Groundwater can respond quickly to precipitation and is the main contribution to streamflow in most catchments in humid, temperate climates. To better understand shallow groundwater dynamics in a boreal headwater catchment, we installed a network of groundwater wells in two areas in the Krycklan catchment in Northern Sweden: a small headwater catchment (3.5 ha, 54 wells) and a hillslope (1 ha, 21 wells). The average wells depth was 274 cm ( range: 70 - 581 cm) and recorded the groundwater level variation at a 10-30 min interval between 18. July 2018 – 1. November 2020. Manual water level measurements (0 - 26 per well) during the summer seasons in 2018 and 2019 were used to confirm and re-calibrate the water level logger results. The groundwater level data for each well was carefully processed and quality controlled, using six data labels. The absolute and relative positions of the wells were measured with a high-precision GPS and terrestrial laser scanner (TLS) to determine differences in groundwater levels and thus groundwater gradients. During the summer of 2019, all wells with sufficient water were sampled and analyzed for electrical conductivity, pH, absorbance, anion and cation concentrations, as well as $\delta18O$ and $\delta2H$. This combined hydrometric and hydrochemical dataset can be useful to test models that simulate groundwater dynamics and to evaluate subsurface hydrological connectivity. We therefore made the data available on https://www.safedeposit.se/projects/82 (Erdbrügger et al., 2022).

## 1. Introduction

In most headwater catchments in temperate climates, streamflow is dominated by groundwater flow, even during rainfall or snowmelt events. In many areas, groundwater is also a major source of solutes, such as nitrogen (Sponseller et al., 2016), dissolved organic carbon (Buffam et al., 2011; Ploum et al., 2020), or mercury (e.g., Eklöf et al., 2015; Munthe and Hultberg, 2004; Vidon, 2012). An accurate simulation of flow pathways and solute transport requires a good understanding of the groundwater flow directions, which depends on the difference in absolute groundwater levels. In temperate climates, groundwater levels are assumed to be related to topography (Condon and Maxwell, 2015; Haitjema and Mitchell-Bruker, 2005; Tóth, 1962; Winter, 1999), but flow directions can change significantly over time and space. They tend to be more slope-parallel during wet periods with high groundwater levels



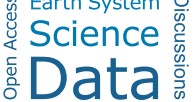

and more stream-parallel during drier periods (e.g., Rodhe and Seibert, 2011; van Meerveld et al., 2015).
The direction can even change from being directed towards the stream to away from the stream (Covino
and McGlynn, 2007; Doering et al., 2007; Payn et al., 2009; Simpson and Meixner, 2013; Ward et al.,
2013; Yu et al., 2013; Zimmer and McGlynn, 2017).

Shallow groundwater levels can increase quickly during rainfall or snowmelt events, and groundwater
level dynamics can vary considerably over distances of just several meters (van Meerveld et al., 2015;
Moore and Thompson, 1996; Myrabø, 1997; Seibert et al., 2003; Tromp-van Meerveld and McDonnell,
2006), thus spatial and temporal resolution measurements of groundwater levels can help to improve
our understanding hydrological systems and their functioning. High-resolution groundwater data are
also useful to better understand the relation between the depth to groundwater and vegetation (e.g.,
Bachmair et al., 2012), to test hydrological models (e.g., Jutebring Sterte, 2016) or methods to simulate
groundwater levels at unmonitored sites or derived variables. For example, several approaches have
been used to quantify hydrological connectivity based on groundwater level data and topography.
Rinderer et al. (2019) estimated groundwater levels for unmonitored sites based on the relation between
the relative groundwater levels and the topographic wetness index (TWI; Beven and Kirkby, 1979), and
Jencso et al. (2009) estimated the duration of hillslope stream connectivity based on the upslope
accumulated area. However, due to the lack of high-resolution groundwater level data in both space and
time, these approaches have rarely been tested for multiple catchments.

Groundwater level data are collected by regulatory or environmental management agencies, such as the
Sverige Geologiska Undersökning in Sweden. Although these large datasets include observations for
many groundwater wells, they mainly contain data on groundwater levels in major aquifers, reflecting
the direct societal importance of these aquifers as a water resource. However, these datasets generally
contain no or only minimal data for shallow unconfined aquifers in headwater catchments. Furthermore,
the spacing of the wells is usually too wide to allow for the calculation of the groundwater flow
directions (but see Fan (2019) and Fan and Schaller (2009), who used data from these types of datasets
to determine the vertical component of groundwater flow across the US).

Groundwater level data have been collected at a high spatial and temporal resolution in a few research
catchments (Table 1). For example, Jencso et al. (2009) measured groundwater levels in 84 wells across
the 17.2 km² Tenderfoot Creek Experimental Forest in Montana, USA, and Rinderer et al. (2014)
measured water level dynamics in 51 wells in the 20 ha Studibach catchment in Switzerland. A few
other studies took manual measurements at many wells (e.g., Moore and Thompson, 1996; Myrabø,
1997), or combined manual measurements with data loggers (e.g., Bonanno et al. 2021). Other studies
have collected high spatial resolution groundwater data for individual hillslopes (for examples see Table
1).One reason that so few high-spatial and temporal resolution groundwater level datasets exist is the
high cost (both time and money) to install and maintain a dense groundwater monitoring network (see
also Retike et al., 2022). Until a few decades ago, water level sensors with an integrated logger or options
for wireless transmission were not readily available. This implied that automatic measurements in



multiple groundwater wells required either multiple data loggers, which were rather expensive, or that sensors were connected to a single data logger by wires, which limited the maximum distances between them or caused other problems (e.g., broken cables and an increased the risk of damage by lightning).

Sensors and data loggers were also more expensive then, so that where multiple wells were available, the measurements were mainly often done by hand, resulting in low temporal resolution data (e.g., Bishop et al., 2011; Seibert et al., 2011; see also Table 1).

In addition to recent advances in data logging, there have also been advances in the development of handheld (but powered) augers (e.g., Gabrielli and Mcdonnell, 2012), which makes it easier to install

multiple wells in a reasonable amount of time. At the same time, drilling rigs and drilling services have become more readily available as well. This has made drilling in remote terrain more practical and the installation of a dense well network easier, though still costly and time-consuming (see section 4.2). To calculate flow directions, the elevation of the wells and their position need to be known accurately (Rau et al., 2019). While this can be done with traditional surveying methods, terrestrial LiDAR

measurements have made it easier to determine the exact positions of groundwater wells in the landscape. In summary, recent technological advances in data logging, drilling, and surveying, have made it easier to collect high-resolution groundwater level data than was previously possible. However, there are still very few public datasets with high temporal and spatial resolution groundwater data due to the time and effort needed to collect, clean and publish the data (see also Retike et al., 2022).

Here, we present a unique dataset with two years of groundwater level data for 54 in a 3.5 ha and 20 wells in a 1 ha study area within the Krycklan catchment in Northern Sweden, where streamflow is dominated by shallow groundwater flow (Laudon et al., 2013, 2021). The shallow aquifer in the study catchment consists primarily of till that is relatively uniform in its lateral extent. Long-term data for precipitation and streamflow make the two years of groundwater measurements even more helpful for

model testing purposes. In addition to the groundwater level data, we also present the results of a sampling campaign during summer 2019 to determine the spatial variability in groundwater chemistry. Groundwater chemistry can be highly spatially variable (e.g., Kiewiet et al., 2019; Penna and van Meerveld, 2019) and are helpful to study groundwater flow pathways and to validate hydrological models or nutrient transport models (e.g., Kolbe et al., 2020). Please see Table 2 for a brief description

of the two datasets, the files for each dataset, and the information contained in each file.



**Table 1: Selected catchment and hillslope scale studies with a large number of shallow groundwater level measurements and reported well densities (number of wells per hectare).**

| Reference | Catchment | Recording wells | | Manual measurements | |
|---|---|---|---|---|---|
| # | | # | #/ha | # | #/ha |
| *Catchment studies* | | | | | |
| Moore and Thompson (1996) | Malcolm Knapp Research Forest, Canada | - | - | 59 | 15 |
| Myrabø (1997) | Sæternbekken catchment, Norway | 4 | | >100 | 0.01 |
| Jencso et al. (2009) | Tenderfoot Creek Experimental Forest, Montana, USA | 84 | 0.5 | - | |
| Rodhe and Seibert (2011) | Östfora experimental catchment, Sweden | - | | 15 | 0.4 |
| Rinderer et al. (2014) | Studibach catchment, Switzerland | 51 | 2.6 | | |
| Bonanno et al. (2021) | Near the Weierbach experimental catchment, Luxembourg. | 22 | 195 | 14 | 125 |
| This study | Krycklan catchment (C6, area A), Sweden | 54 | 15 | - | |
| *Hillslope studies* | | | | | |
| Tromp-van Meerveld and McDonnell (2006) | Panola Mountain Research Watershed trenched hillslope | 29 | 29 | 135 | 135 |
| Vidon and Smith, (2007), Vidon (2012) | Scott Starling Nature Sanctuary site/Riparian Zone | - | | 14 | 2.3 |
| Haught and van Meerveld (2011) | Malcom Knapp Research Forest, Canada | 18 | 819 | - | |
| Bachmair, Weiler and Troch (2012) | Southern Germany | 90 | 4.3 | - | |
| This study | Krycklan catchment (S-transect, area B), Sweden | 20 | 20 | - | |
| Bishop et al. (2011), Seibert et al. (2011) | Gårdsjön Covered Catchment | 3 | 4.8 | 34 | 54 |



Table 2: Name and short description of the datafiles included in the two datasets

| Dataset | File names | Short description | Contents described in |
|---|---|---|---|
| 1 | Krycklan_gw_wells.csv | well position (x,y,z), height of correction factors, etc. | Table 4 |
| 1 | Krycklan_gw_levels.csv | Groundwater levels from manual measurements and loggers, tube above ground, etc. | Table 5 |
| 1 | TSL_registration_report_[A/B].rtf | TSL scan registration report | Section 3.3 |
| 2 | Krycklan_gw_sampling.csv | Sampling information for groundwater chemistry | Table 7 |
| 2 | Krycklan_gw_chemistry.csv | Chemistry data from groundwater sampling | Table 8 |
| 2 | Field_protocol.csv | Field protocol for sampling of groundwater | Section 5 |
| 2 | Lab_analysis_description.pdf | Information on the laboratory analyses | Section 5 |


## 2. Description of the Study Areas

The Krycklan catchment is located in Northern Sweden, about 60 km inland from Umea. The research catchment is used to collect long-term data on hydrological, climatic, biogeochemical, and environmental variables (see Laudon *et al.* (2013) and Laudon et al. (2021)). The region has a cold temperate humid climate, with on average 167 days of persistent snow cover (1981–2020 period) but this duration has been declining in recent years (Laudon et al., 2021). The mean annual temperature (1981–2010) is 1.8° C; with a mean temperature in January of -9.5 °C and a mean temperature of 14.7 °C in July). The mean annual precipitation is 614 mm/y and the mean annual runoff is 311 mm/y (Laudon et al., 2013, 2021). The landscape in Krycklan is influenced by the last glaciation, which formed a hilly landscape with rock outcrops, moraine ridges, and valleys that have thicker till deposits and fluvial deposits from the Vindeln river. The highest postglacial coastline crosses the study area at an elevation of approximately 257 m above mean sea level (amsl)(Laudon et al., 2021).

The two study areas where the wells were installed are located in the core area of the Krycklan research catchment (Figure 1). One area is located in what is called sub-catchment 6 (110 ha) in other studies (e.g. Laudon et al., 2021). This is referred to study area A here. The other area is located near what is called the S-transect in sub-catchment 2 (12 ha). This area is hereafter referred to as area B. The elevation



of the study areas ranges from approximately 250-270 m amsl. The landscape consists primarily of pine (Pinus sylvestris) and spruce (Picea abies) forest (87%) and mires (9%). Both study areas are located within the zone that has been protected since 1922 and where forestry activities were limited. However,

the area was subject to ditching around the beginning of 1900 (Laudon et al., 2013, 2021). The soil in the study areas is comprised of till with occasional clusters of large boulders (see Ivarsson and Johnsson, 1988). The soil is very thin close to rock outcrops near the catchment ridges and up to 6 m towards the streams. The saturated hydraulic conductivity of the soil declines rapidly with depth below the surface (at least for the upper meter of the soil (Ameli et al., 2021; Bishop et al., 1990)).

The shallow aquifers in the study catchment consist mostly of till that is relatively uniform in its lateral extent. The groundwater tables are generally shallow (< 6 m from the surface, and in most locations < 2 m). Shallow groundwater flow is the main source of streamflow (Ledesma et al., 2018; Lyon et al., 2012) and is driven by the topography (Leach et al., 2017; Ploum et al., 2020). Kolbe et al. (2020) found that just below the water table, the water age suddenly seems to increase by several decades.

A particular advantage of obtaining high density and high-resolution groundwater data in the Krycklan catchment is the abundance of other data (e.g., precipitation, streamflow, vegetation, soil, etc.; Laudon et al., 2021, 2013), as well as atmospheric data from the Integrated Carbon Observatory System (ICOS) Svartberget station (ICOS, 2021), which is located close (<500 m) to the study areas.

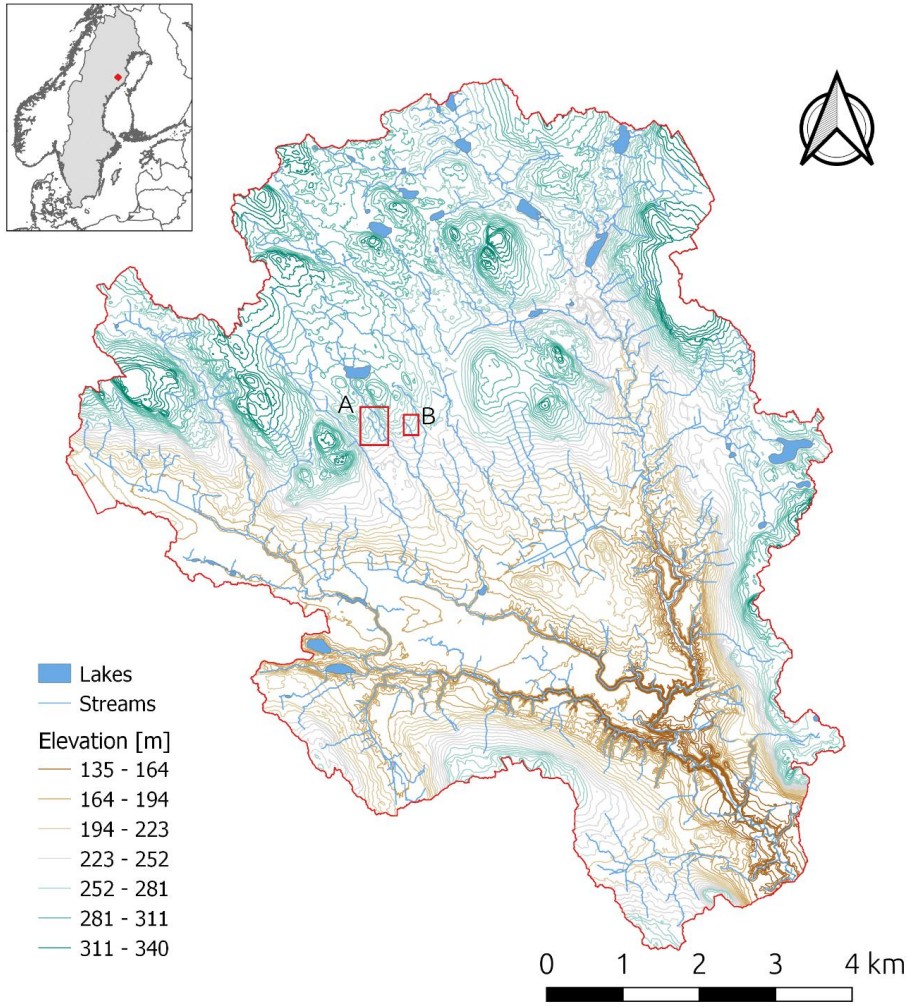

**Figure 1: The Krycklan catchment and and the location of the two study areas (red squares, A (in C6) and B (s-transect in C2)) with the networks of groundwater wells. The inset shows the location in of the Krycklan catchment in Northern Sweden. See Figure 2 for a more detailed map of the study areas.**

### 3. Groundwater wells

### 3.1 Well network design

The locations for the wells were chosen based on the LiDAR-derived digital elevation model (DEM), with a closer well spacing in areas where we either expected very stable or very variable flow directions/gradients (Erdbrügger et al., 2021). To determine groundwater gradients (i.e., flow directions), the wells were located in triangles of different sizes: 5 m, 10 m and 20 m (Figure 2). In the field, the planned positions of the wells were identified with a handheld GPS and the elevation and vegetation maps. Not all wells could be installed as planned. For some locations, the position had to be adjusted due to the presence of trees and big boulders. In addition, the access required for the drill rig

(see section 3.2) meant that the location of some of the wells had to be moved. However, in most cases the wells could be installed within 5 m of the pre-determined positions based on the DEM.

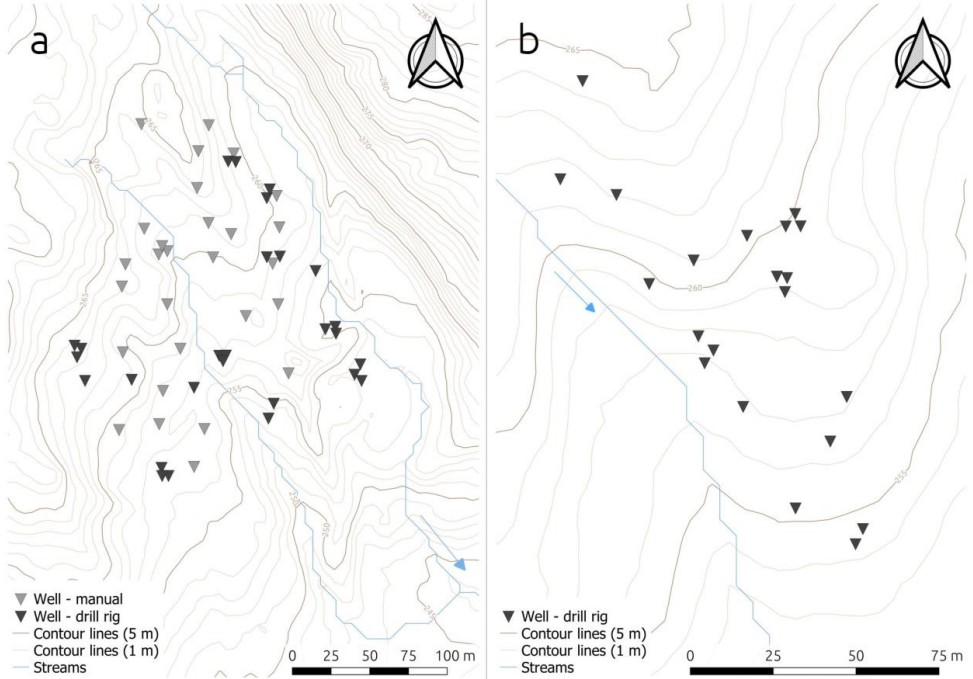

**Figure 2: Maps of the two study areas (A (a) and B (b)) with the location of the 75 wells (triangles) that were either augured by hand (manual, in light gray) or installed by the drill rig (in black). See Figure 1 for the location of the two study areas within the Krycklan catchment.**

### 3.2 Well installation

We installed 27 wells with a Cobra™ Petrol-Driven Drill and Breaker between May and October 2018 (all in area A) and 48 wells (27 in area A and 21 in area B) with a drill rig between February and March 2019 (see Figure 3). The drill rig was only used in winter, when the ground was frozen and covered by a thick snowpack, to minimize the impact of the heavy machinery on soil and vegetation. The wells consist of fully screened PVC pipes with an outer diameter of 5.0 cm and an inner diameter of 3.7 cm. A filter sock was placed over the entire screened length of the pipe to limit the entry of particles into the well.

The required well depths were estimated based on the "Depth to Water" index (Murphy et al., 2009) that was calculated based on the DEM but were adjusted based on local groundwater level observations. Many of the wells needed to be deeper than suggested by the index. For the installations during winter, the depths were based on groundwater depths measured in the summer. We tried to install the wells at least one meter deeper than the lowest observed groundwater level in nearby wells, but in some cases, the target depths could not be reached due to boulders or other obstacles. Some wells were not sufficiently deep to measure the groundwater level during the driest periods. This was particularly the



case for the wells located furthest away from surface water bodies. The average depth of the wells was
274 cm (standard deviation: 113 cm; range: 70 - 581 cm). The depth of the wells installed in summer
(with Cobra) was generally less (average 193 cm) than for the wells installed in winter with the drill rig
(average: 316 cm).

The height of the top of the wells above the ground was measured several times after installation, as
well as three occasions in 2018 and two occasions in 2019. A marker on the pipe ensured that the height
was always measured on the same side of the pipe. These measurements of the stick-up were used to
determine water level below to the ground surface from the absolute levels.

**Table 3: Estimated costs for installing the well network**

| What | Notes | Cost |
|---|---|---|
| Well installation with machinery | February-March 2019, drilling company in charge of 52 wells with depths ranging from 2 – 5 m (including installation of fully screened PVC pipes) | ~15 500 EUR (~300 EUR per well) |
| Well installation manually | Summer season 2018, 3 student interns for 3 months (half-time) with Cobra | (~ 5 800 EUR for a new cobra not including well installation equipment, payment of staff depends on local wages) |
| Water level loggers | Depending on cable length. In total 75 wells equipped Odyssey capacitance water level loggers (Dataflow Systems Ltd, 2021) | ~ 210 EUR per logger |
| PVC pipe and filter sock | Fully screened PVC groundwater tubes, filter sock, etc. | ~10-20 EUR per well |



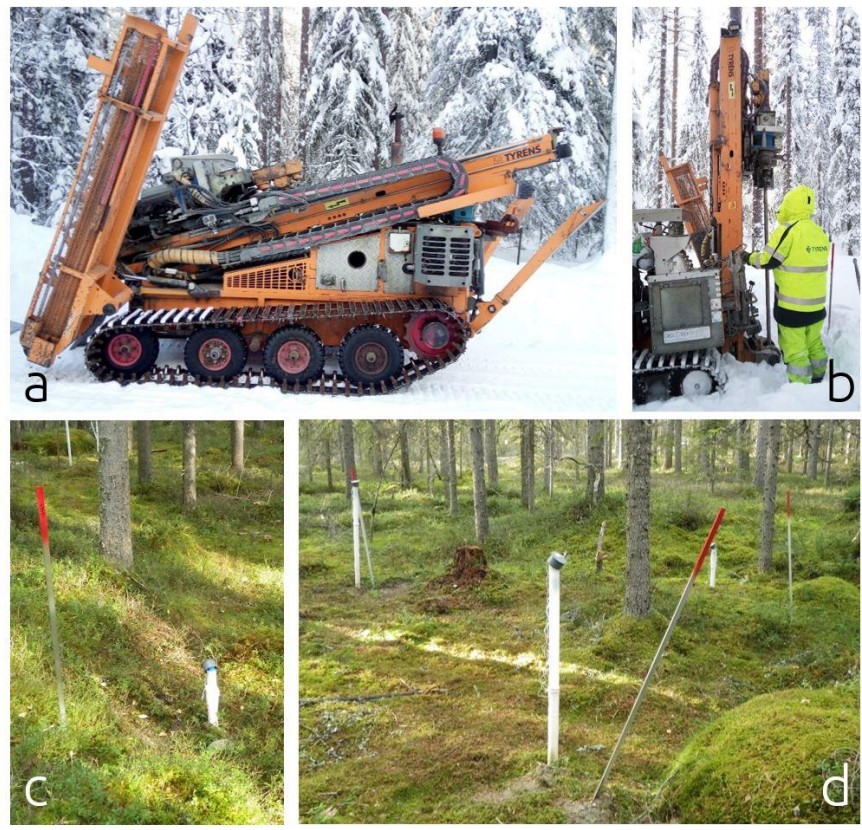


**Figure 3: ((a) and (b)) Well installation in winter 2019, picture of an installed well (with ~ 35 cm of pipe above ground), and ((c), (d)) closely positioned wells that form triangles (length of pipe above ground appr. 50 cm in panel (c) and 110 cm in panel (d)). The gray caps were placed over the loggers for protection. The aluminum poles with the red paint next to the wells were used to determine the location of the wells when snow covered the ground.**


### 3.3 Well geo-referencing

The position of the wells was determined with a high-precision GPS after installation. All wells were scanned with a terrestrial laser scanner (Trimble TX8) in May 2019 to more accurately determine their vertically and horizontally position relative to each other. About 68 single scans (39 in area A and 29 in

area B) were done at a distance of about 20 m. The scan resolution at 30 m from the scanner was one point every 11.3 mm (Drive and Trimble Inc., 2017). The generated point clouds were combined into one large point cloud following the procedure proposed by the Ljungberget Remote Sensing Laboratory at the Swedish University of Agricultural Sciences (Bohlin and Nyström, 2019). The wells were manually identified in the combined point cloud, and the upper end of the tube was taken as the reference

point (Figure 4). The relative positions of the wells are given in the Krycklan_gw_wells.csv file. The registration reports are given in the TLS_registration_area[A/B].rtf-files); the full scan data are available via the Krycklan database (Lindgren, 2021).





The positions of the well tops were exported as an ESRI shape-file. We then used a similarity transformation (only rotations and x and y offsets, no scaling) to georeference the well positions (x and y coordinates only). As orientation points for the similarity transformation, we used the locations of four wells measured in the field with the high-precision GPS. Since the scan was already level in the horizontal direction (using the TLS internal leveling), we adjusted the z-offset by the GPS z-position of one of the wells to obtain a general offset for all wells. The procedure was carried out separately for the two study areas (A, B) (Table 3).

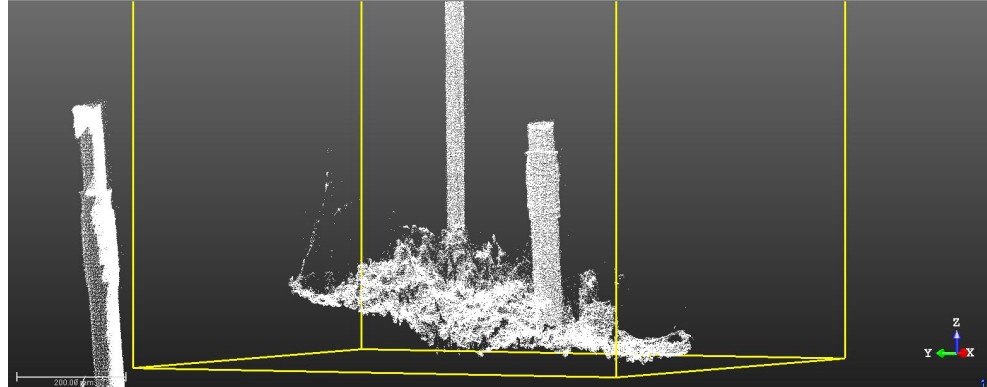

**Figure 4: Example of the point clouds for manually identified wells. The well (on the right and in the foreground on the left), filter sock, and the aluminum pole (in the middle, top cut off) marking the well and surrounding vegetation are clearly visible in the point cloud.**

We used the elevation of the well top (i.e., top of the PVC pipe) to determine the water level (see sections 4.2 and 4.3). The position of the well tops may have shifted slightly over the measurement period due to freezing and thawing of the ground. Soil heave in the Krycklan catchment can be several centimeters over the frost season (Bergsten et al., 2001). We, however, assume that this effect was minor for the wells (and thus the well tops) because the pipes reached well below the average freeze/thaw line, which is located at -19 cm in the Krycklan catchment (Panneer Selvam et al., 2016). Thus, although the relation of the tube top to the soil surface may have changed slightly over time due to soil heave, as well as trampling, we assume that the relation of the well tops to each other remained the same.

## 4. Dataset 1: Groundwater levels

### 4.1 Dataset structure

The groundwater level dataset (Dataset 1) consists of two files. One file (Krycklan_gw_wells.csv) provides a description of each well (Table 4) and the other file (Krycklan_gw_wells.csv) provides the time series of the actual measurements and the calculated water level in meters above mean sea level (m



amsl) for each well (Table 5). The measurements and data processing steps to obtain the time series of
the groundwater levels are described in the following sections.

**Table 4. Structure of the groundwater well location data file (Krycklan_gw_wells.csv) and description of the column names**

| Column name | Full title | Description |
|---|---|---|
| Well | Well_name | Name of well |
| Well_field | Well field name | Name of the well used in the field originally |
| X | X-Coordinate | X coordinate [m] as extracted from TLS Datum: EPSG:3006 - SWEREF99 TM |
| Y | Y-Coordinate | Y coordinate [m] as extracted from TLS Datum: EPSG:3006 - SWEREF99 TM |
| Z_abs | Z-tube top | Absolute elevation of pipe top above mean sea level extracted from TLS [m amsl] |
| Depth | Well depth | Depth of the well below the ground surface at the time of installation [m] |
| intercept | Recalibration offset | Recalibration offset calculated based on the linear regression between the manually measured water levels and the logger derived data, used to correct the logger measurements (only applied correction values) |




**Table 5 Structure of the groundwater level data file (Krycklan_gw_levels.csv) and description of the column names**

| Column name | Full title | Description [unit] |
|---|---|---|
| Well | Well_name | Name of well |
| datetime | Date and time | Date and time (CET) of the measurement<br>Format: DD/MM/YYYY hh:mm:ss (no energy saving time) |
| tube | Length of the pipe above ground | Length of the PVC pipe above the ground [cm] as measured during field season (i.e., stick-up). Values are assumed to remain constant between measurements. |
| mnl_level | Manual water level | Groundwater level [m amsl], calculated by subtracting the manually measured distance between the water level and the top of the pipe (man_level) from the from the absolute elevation (Z_abs) |
| Log_level_uncorr | Water level below top of well (uncorrected) | Water level from logger [cm from the top of the tube], calculated based on the logger length and string length (if logger was suspended inside the well). Note that this is the level before the re-calibration with manually measured water levels |
| Log_level_corr | Absolute water level (corrected) | Water level after correction and subtraction from the absolute elevation (Z_abs) [m amsl] |
| Class | Data classification | Data point flagging as valid, outlier, recovery, etc. (see Table 6) |
| Class_mnl | Manual Data classification | Manual data point flagging based on field observations (valid, outlier) |

### 4.2 Manual water level measurements

The distance between the top of the well and the water level (*man_level*) was manually measured at a
weekly to a bi-weekly interval in July, September and October 2018 and between May and September
2019. On average, the depth to the water level could be manually measured 14 times (range: 0-26). For
the shallow (< 1 m) groundwater levels, we usually used a bubbler to measure the distance between the
top of the well and the water level. For the deeper wells we used either a water level plopper ("kluk lod"
in Swedish) or an acoustic water level sounder. For more detailed information on these measurements,
see Appendix A. The distance to the water level was noted directly in a spreadsheet and on paper for
cross-referencing after the fieldwork. These data were screened visually and checked for plausibility.
Data points that deviated strongly from the expected range were double-checked based on field notes,



meteorological data and data from nearby wells. Data entries outside the expected range that could not
be verified by any other source were classified as outliers. In total, five manual measurements that
deviated more than 1 m from the expected value were excluded and are assumed to have been entered
incorrectly in the datasheets.

### 4.3 Continuous water level measurements

#### 4.3.1   Initial logger calibration, installation and maintenance

For continuous water level measurements, we installed capacitance water level loggers (Dataflow
Systems PTY Limited, Christchurch, New Zealand) in 74 wells. The length of the cable of the water
level loggers was based on the depth of the well and the estimated changes in water level. Although we
used loggers with different housing lengths and cable lengths, they all function similarly (see Appendix
A).

All loggers were calibrated according to the instructions provided by the supplier (Dataflow Systems
PTY Limited, 2012) prior to field installation. In short, two points (at 20 cm and 140 cm from the lower
end of the weight at the end of the cable; see Figure A 1) were marked and the logger was suspended in
a sealed PVC pipe filled with water from one of the groundwater wells so that it reached exactly the
mark on the cable. For each position, the raw measurement values were noted after an acclimatization
phase of about half a minute or the values had stabilized. This two-point linear calibration was used to
convert the raw sensor values to distances (in cm) above the bottom of the sensor.

The loggers were set to record at a 10 min interval, except in the first measurement week when it was
15 min, and in winter 2018/2019 when it was set to 30 min to avoid filling the memory (and overwriting
the data) during the winter period. The data were downloaded three or four times during the field season
(May to October). Loggers that did not record any data or only recorded data for part of the time were
inspected and usually reinstalled the same day or the following day.

For the manual water level measurements, we had to take the loggers out of the wells. We tried to time
this so that it would not coincide with the measurements (i.e., we aimed to do this within the 10 min
interval between measurements). The loggers were inspected visually for any disturbances, such as
biofilm or dirt on the cable, kinks, or obstructions, and were cleaned when necessary. We also measured
and adjusted the logger string lengths if the groundwater level had fallen or was expected to fall below
the deepest point of the sensor or if it was expected to rise above the logger body during snowmelt.

The groundwater level (*Log_level_uncorr*) in cm below the top of the well was calculated by subtracting
the water level measured by the logger (in cm from the bottom of the logger weight) from the sum of
the length of the logger body, cable and weight length, and string length (see Figure A 1). These water
level time series were inspected manually for incongruences. For five wells, the string length
information was accidentally not recorded for all time periods (after adjustments due to very high or
very low water levels in the well) so that the groundwater level below the top of the well could not be
calculated; these data points were marked (*Offset* flag, see section 4.4).



### 4.3.2 Logger level data correction (re-calibration)

When analyzing the data, it became apparent that there was a systematic offset between the logger data (*Log_level_uncorr*) and the manual measurements (*man_level*) (Figure 5 (a) and (c)), with the logger-based water levels being systematically higher (i.e., closer to the surface) than the manual measurements. In some cases, they were above the surface, although we only observed flooding for two of the wells (i.e., only for these two wells did we expect the water levels to rise above the ground surface at some

point in the year). We deemed the manual measurements to be more reliable and assumed that the shift was due to a systematic error in the calibration of the loggers or calculation of the logger water levels. We can exclude pipe length (which would have been an individual error for each well) and string length (the shift also appears for loggers that were not attached to strings) as the source of the systematic error and, therefore, assume that the shift was due to a systematic offset in the calibration of the loggers before

field deployment.

We corrected the logger data in a two-step process based on the assumption that we only needed to correct the vertical shift in the logger data. We first determined the linear regression between the manual measurements and logger data, with a slope of 1. We then defined points more than 5 cm from our linear regression as outliers to account for potential errors in the manual water level measurements. After

excluding these outliers, we determined the new linear regression between the manual measurements and logger-derived water level data, again with a slope of 1, and calculated the offset (see two examples of the correction in Figure 5). The mean value for this correction (i.e., *intercept*) was -9.91 cm (standard deviation: 6.74 cm; range: -28.1 - -0.05 cm). We then used the offset (*intercept*) to correct the logger data. Thus, effectively, we lowered the logger data to match the manual water level measurements. This

fixed the issues with the water levels rising above the surface for all wells for which this was not observed. The correction (*intercept*) was only calculated for the wells for which there were at least two valid manual water level measurements (i.e., two points that were not considered outliers after the first step). The data were not corrected for 18 of the 74 wells with loggers because there were insufficient valid manual water level measurements to calculate the correction (i.e., *intercept*).

The final absolute groundwater level (*Log_level_corr*; in m asl) was then calculated by adding the correction factor (*intercept*) to the uncorrected logger level (*log_level_uncor*) and the elevation of the well top (*Z_abs*): *Log_level_corr = Z_abs – (Log_level_uncorr+intercept)*.

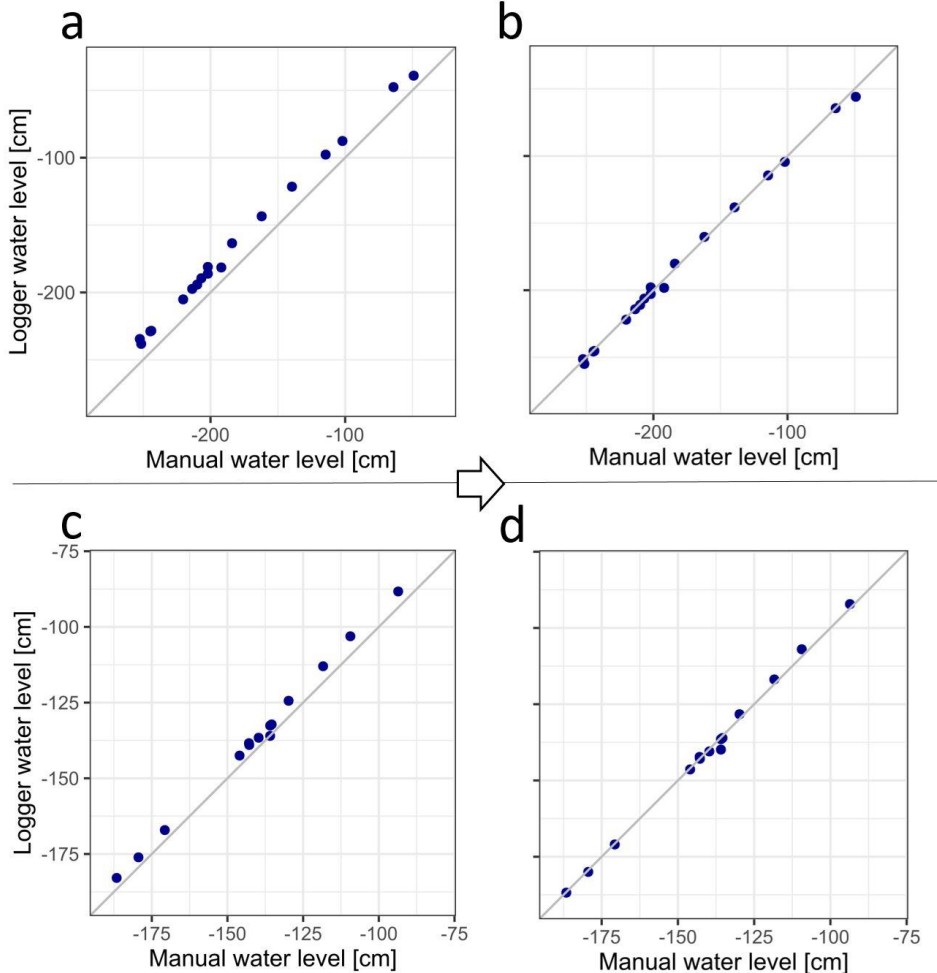

**Figure 5: Relation between the manual and logger water level measurements in cm below the top of the well before ((a) and (c)) and after ((b) and (d)) recalibration of the logger for well A12 ((a) and (b)) and well B18 ((c) and (d)). The correction factor (i.e., intercept) was -16.8 cm for well A12 and - 3.9 cm for well B18.**

### 4.4 Data Flagging

Observed outliers or discontinuities in the water level time series were classified into six different categories (Table 6; Figure 6). The flagging allowed us to keep all data points in the record and re-evaluate the classification if necessary.





**Table 6: Classification of data points**

| Class | Description |
|---|---|
| Valid | No known or apparent reason to indicate that the measurement is not valid. This is the default classification for all data points |
| Outlier | Known outliers (during pumping) and unknown outliers (>10 cm drop in ≤10 min) |
| Recovery | Recovery of water levels after pumping (0-12 h after re-introduction of the logger into the well) |
| Censored | Uncertain values due to low water levels (< 5 cm above logger bottom) or water in lower well ends (lower 5 cm of unscreened pipe) |
| Strange | Snowmelt curve with sudden breaks (probably related to a rapid decrease in the electrical conductivity of the water) |
| Offset | String length or tube length were unkown. The relative changes in water level are correct but the absolute level could not be calculated. |


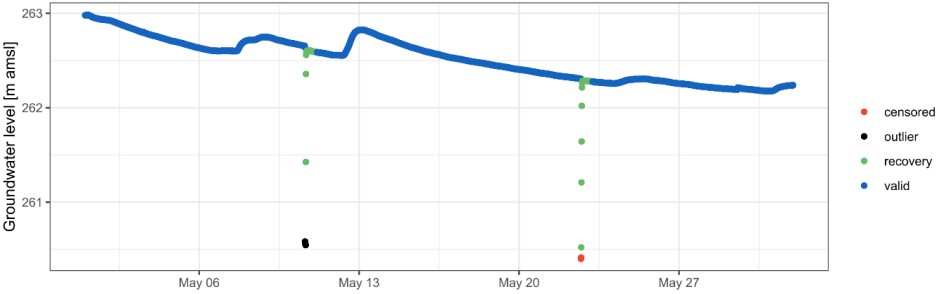

**Figure 6: Example of a groundwater time series (well A12, May 2019) affected by two pumping events, with identification of the outliers (black) during the time that the logger was outside the well (while the pumping took place) and the recovery period (green). Water levels below the threshold of 5 cm above the logger end are classified as censored (red).**


To download the loggers, take manual water level measurements, or purge the wells for cleaning and sampling (see description of Dataset 2), the loggers were taken out of the wells. This usually took a short

time, and therefore, the loggers were not stopped. Most of the measurements taken during these periods were significantly lower than the measurements taken before or afterwards and are therefore classified as *outlier*s. To find these outliers, we used a filter based on the changes in the water levels. Assuming that groundwater levels generally do not drop abruptly (i.e., the recession is smooth), we used a threshold value of a > 10 cm drop in groundwater level within 10 min to find outliers, which were then manually

investigated to ensure correct identification.

Data points collected after the re-introduction of the logger into the well after well purging were classified as *recovery* to mark the recovery of the water level within the well. To be certain that





equilibrium had been reached, all data points within 12 hours after the re-insertion of the logger to the well were classified as recovery. Where the recovery time appeared to exceed the 12-hour time span,

we extended the classification to 24 hours after re-introduction of the logger.

When the groundwater level was close to the weight at the end of the sensor cable, the recorded data often showed sudden jumps, suggesting a low accuracy of the measurements. To eliminate this problem and the problem of standing water in the very bottom of the well (which was not screened), we classified all points that were less than 5 cm above the logger end as *censored*.

For wells five of the 74 wells (A21, A9, A4, B1, and B6), we observed a continuous drop in the water level between 21.04.2019 and 30.04.2019 (see examples in Figure 7). This was the peak snowmelt period for which we expect the groundwater level to rise rather than drop. The start of the drop in the water level differed for the wells but it ended suddenly for all five wells on 30.04.2019, when the loggers were removed from the wells to download the data. After the re-introduction of the loggers in the wells,

the measured water levels were several centimeters higher than before. The logger string or tube lengths were not adjusted during this period and the sudden change in the recorded water level can therefore not be related to errors in these measurements. The wells for which this strange behavior was observed were not located in one region or characterized by a particular topographic position. These were all located close to other groundwater wells (within ~ 5 m), for which we did not observe such a change. The data

from these wells during this period are therefore flagged as *strange*.

Because the recorded time series after the re-introduction of the loggers to the well seems to agree with the water levels observed in the days before the sudden drop, we expect that the lowering of Electrical Conductivity (EC) of the water in the wells due to the infiltration of low EC meltwater caused this change. Odyssey water level logger recordings can be sensitive to a large change in EC (Larson and

Runyan, 2009). Most probably, the removal of the logger from the well and re-introduction stirred the water inside the wells and led to a mixing of the snow meltwater and older groundwater. The alternative explanation of a film (biological or other) on the cables does not correspond with our field notes. Only in two cases was a film observed on the sensor cables, but for these sensors, the logged values seem to be normal during this period.


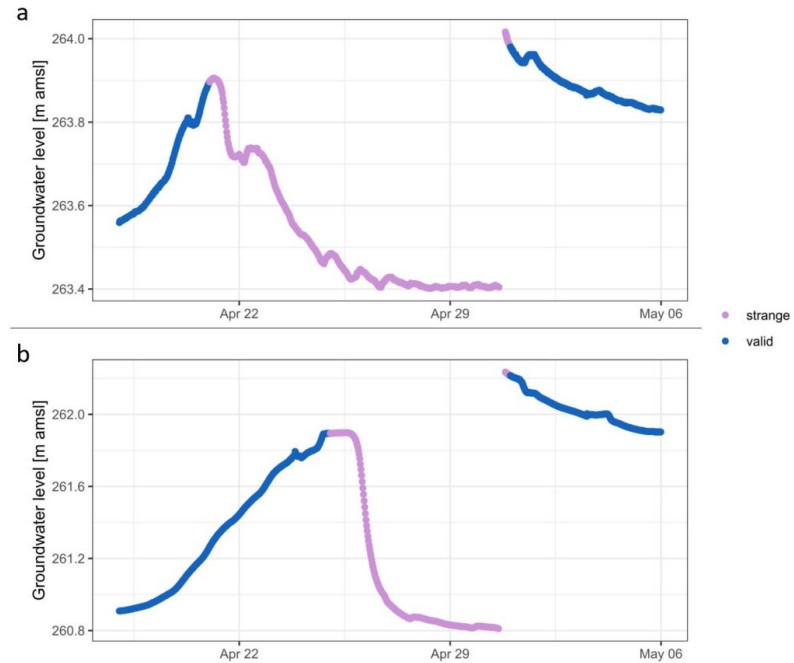

**Figure 7: Examples of strange drops in the measured water level during the 2019 spring melt period for two wells (well A4 (a) and well A21 (b)).**

## 5. Dataset 2: Groundwater Chemistry

**5.1 Dataset structure**

In summer 2019, a groundwater sampling campaign was undertaken to obtain spatially distributed information on groundwater chemistry. The resulting groundwater chemistry dataset (Dataset 2) consists of four files. One file (Krycklan_sampling.csv) provides a description of each sample (Table 7) and another one (Krycklan_chemistry.csv) the laboratory results for each sample (Table 8). The third file

contains the field protocol (Field_protocol.csv), and the fourth file (Lab_analysis_description.pdf) provides additional details on the laboratory analyses. The sampling and data analyses are described in more detail in the following sections.



**Table 7: Description of groundwater sampling data (file: Krycklan_sampling.csv)**

| Column name | Full title | Description [unit] |
|---|---|---|
| Well | Well name | Name of the sampled well |
| sample | Upper [1] or lower [2] sample within well | Upper (first [1]) or lower (second [2]) sample from well. If only one sample was taken, it was also noted as upper (first [1]) sample |
| Depth | Depth of sample | Sample depth (midpoint of the sample range, 40 cm) in absolute elevation [m amsl] |
| date | Date of sample | Date the sample was taken [DD/MM/YYYY] |
| sample_num | Sample number | Sample number assigned for SLU laboratory analysis |
| wl | Water level | Water level measured manually directly before sample extraction [m amsl], corresponds to mnl_level in Dataset 1. |
| quality | Sample quality | First impression in field of sample quality (turbidity/suspended sediment, air intrusion in sampling tube, etc.) [g] = Good, [d] = Doubtful (possible influence), [b] = Bad (influence on water sample) |




**Table 8: Description of groundwater chemistry data (file: Krycklan_chemistry.csv), see Lab_analysis_description.pdf -file for more information on the laboratory analysis.**

| Column name | Full title or description | [units] |
|---|---|---|
| sample_num | Sample number assigned for SLU laboratory analysis | Sample number assigned for SLU laboratory analysis |
| d18O | $\delta^{18}O$ | [‰ relative to VSMOW] |
| d2H | $\delta^{2}H$ | [‰ relative to VSMOW] |
| EC | Electrical conductivity EC | [µS/cm] |
| pH | pH | [-] |
| EC_field | Electrical conductivity measured in the field | µS/cm] |
| pH_field | pH measured in the field | [-] |
| absorb_254 | Absorbance at 254 nm | [A/cm] |
| absorb_365 | Absorbance at 365 nm | [A/cm] |
| absorb_420 | Absorbance at 420 nm | [A/cm] |
| absorb_436 | Absorbance at 436 nm | [A/cm] |
| Flu | Fluoride | [mg/L] |
| Cl | Chloride | [mg/L] |
| SSO4 | Sulfate-$SO_4$ | [mg/L] |
| PPO4 | Phosphate-$PO_4$ | [µg/L] |
| NNH4 | Ammonium-$NH_4$ | [µg/L] |
| NNO3 | Nitrate-$NO_3$ | [µg/L] |
| Na | Sodium | [mg/L] |
| NH4 | Ammonium | [mg/L] |
| K | Potassium | [mg/L] |
| Mg | Magnesium | [mg/L] |
| Ca | Calcium | [mg/L] |

### 5.2 Well purging prior to sampling

To ensure that the water samples were representative of the local groundwater, we did two rounds of purging with a peristaltic pump (see Figure A 2) between the middle and the end of May 2019. The loggers were removed for the duration of the pumping. We measured the depth to the water table prior to purging. During each round, the wells were either pumped dry or a volume of at least three times the





well volume (see Appendix B) was pumped out to ensure full replacement of the well water with "fresh"
groundwater. The pumping also removed the sediment and other particles from the wells.

### 5.3 Sampling

The groundwater sampling was done between July 19th and 31st 2019. Total precipitation during this
period was relatively low (28.16 mm in total, max. daily precipitation 13.49 mm). The sampling
procedure and field protocol used for the campaign followed the Svartberget research station and
Swedish University of Agricultural Sciences (SLU) standard procedure for groundwater sampling (see
also the Field_protocol.csv-file).

A custom-made straddle packer system with inflatable rubber tubes (see Appendix B, Figure A 3)
allowed us to isolate a specific part of the well and sample water from a roughly 25 cm long interval.
We aimed to obtain two samples per well: one sample just below the groundwater table and another
sample from the deepest part of the well. Because many wells were shallow and groundwater levels
were low at the end of July, there was insufficient water for two samples for many wells. In these cases,
we took only one water sample from the deepest point of the well.

Prior to taking the actual sample, the tubes were flushed with ~2 L of well water to reduce cross-
contamination of samples and the sample bottles were rinsed three times to avoid contamination. All
bottles were filled to the top, without air bubbles. Additionally, we measured the electrical conductivity
(EC) and pH in the field with a pH/Cond 3320 sensor (Xylem Analytics Germany GmbH). The pumping
was done slowly to not draw water from above, allow for recharge, and avoid excessive aeration of the
samples. However, in some cases, the recharge of the wells was so slow that even the lowest pumping
rate was too high, and the water level in the well dropped, or the well was pumped dry.

### 5.4 Lab analyses

The samples were analyzed for: EC, pH, absorbance at 254 nm, 365 nm, 420 nm, and 436 nm, anion (F,
Cl, S-SO$_4$) concentrations, nutrient (P-PO$_4$, N-NH$_4$, N-NO$_3$) concentrations, and stable isotopes ($\delta^{18}$O
and $\delta^2$H) in the laboratory of SLU, Sweden. The cation (Na, NH$_4$, K, Mg, Ca) concentrations were
analyzed at the Hydrogeological Laboratory of the TU Bergakademie Freiberg, Germany. A detailed
description of the lab procedures is given in the file: Lab_analysis_description.docx.
.

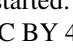

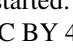

**Figure 8: Precipitation (first plot, blue bars), temperature (daily average, second plot, red line) and classified groundwater levels for two wells (A12 third plot, B18 fourth plot) between June 2019 and September 2020. See Table 6 for further information on the classification of the data points and Figure 6 for the legend.**


## 6. Example results

### 6.1 Groundwater level data

Figure 8 shows the groundwater level time series for two selected wells (A12 and B18) and highlights the quick response to the snowmelt period and rainfall events. Thanks to the relatively high resolution

of the groundwater data, the immediate response of groundwater levels to specific rainfall events is clear. Interesting dynamics like daily groundwater level variations due to snowmelt, with peak water levels occurring during the early evening (Figure 9 (a)) and due to evapotranspiration with groundwater levels dropping during the day and stabilizing at night (Figure 9 (b)) can also be observed for many wells (cf. Kirchner et al., 2020).

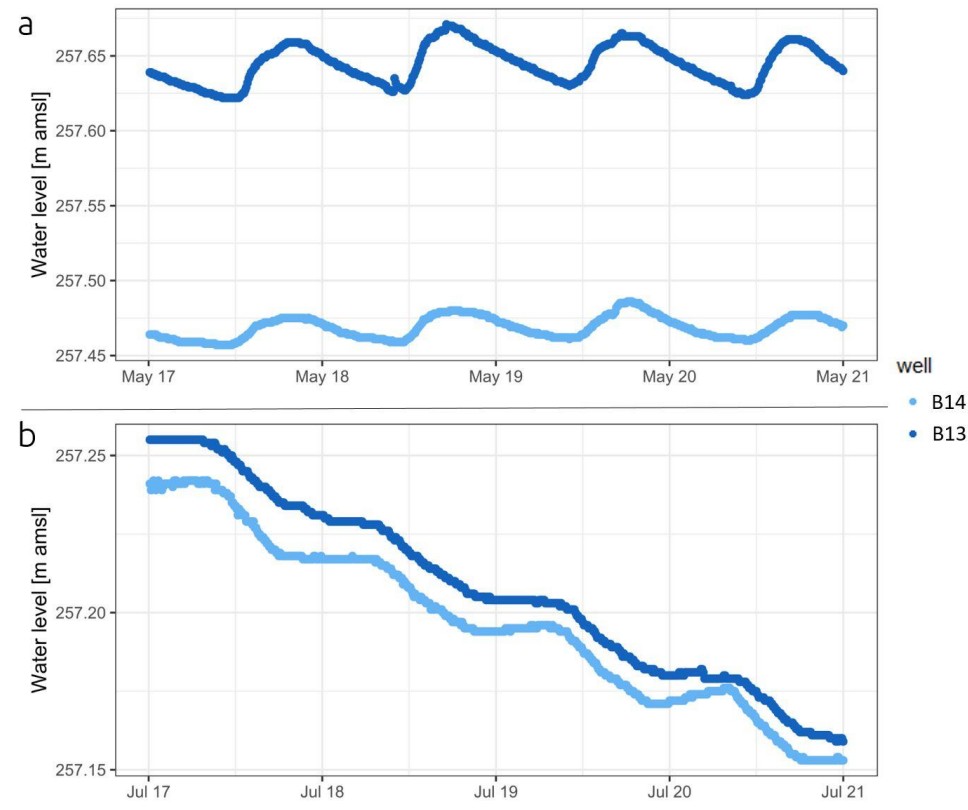


**Figure 9: Example time series of diurnal variation groundwater level for two wells during the late snow melt period in May 2020 (a) and the summer (July 2020) when diurnal variations are caused by evapotranspiration (b).**



### 6.2 Groundwater Chemistry

The concentrations of the anions, cations, and nutrients in the groundwater are comparable to those reported from other measurements in the Krycklan catchment, but the spatial variation in the concentrations was very spread (cf. Kiewiet et al., 2020), with the coefficient of variation ranging from 2.5 % (for $\delta^{18}O$) to 238.8 % (for F) (see examples Figure 10 and Figure 11). In general, the concentrations for the deep groundwater samples differed from those of the shallow groundwater

samples (e.g., Figure 10), but the differences were not statistically significant. The isotopic composition of the groundwater was also variable, with the $\delta^{2}H$, for example, varying by 1.25 ‰ for the shallow groundwater samples (Figure 11).

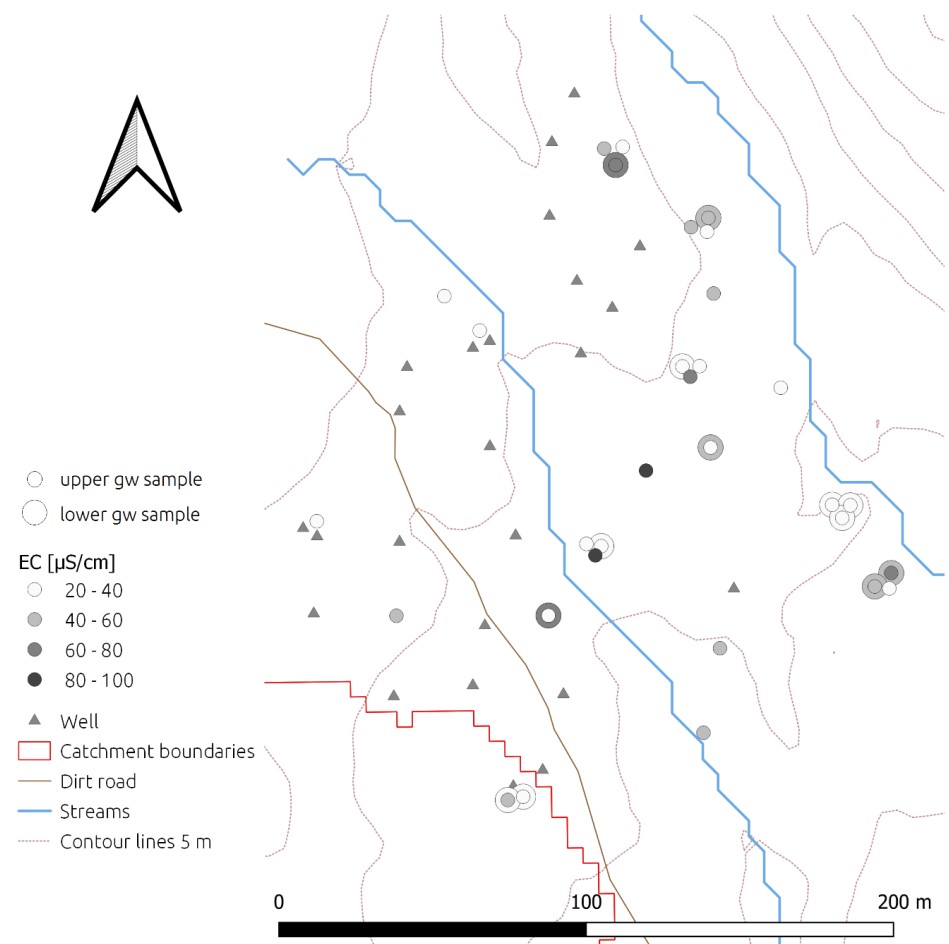

**Figure 10. Map of the electric conductivity (EC) of shallow and deep groundwater in the study area A (C6 catchment).**
**Values for shalow groundwater samples are indicated with a small circle and values for deep groundwater with the larger circle. Thus where two circles are shown, the outer circle represents the deep groundwater and the inner circle the shallow groundwater.**

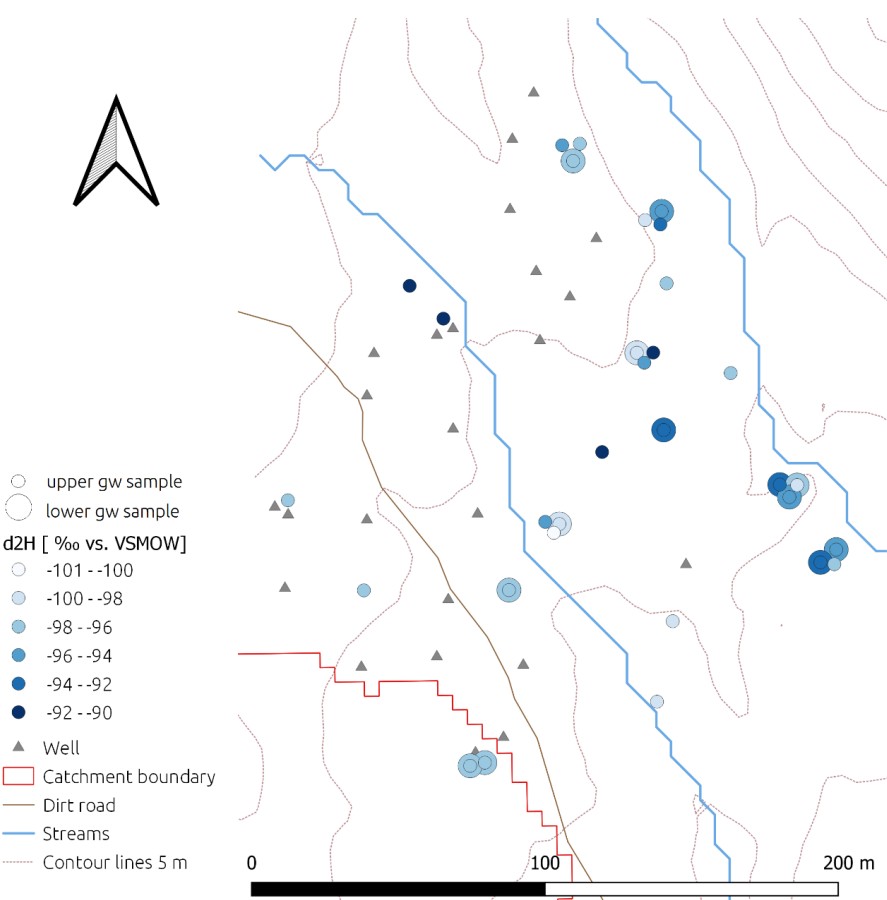

**Figure 11: Map of δ²H of shallow and deep groundwater in study area A (the C6 catchment). Values for shallow groundwater samples are indicated with a small circle and values for deep groundwater with the larger circle. Thus where two circles are shown, the outer circle represents the deep groundwater and the inner circle the shallow groundwater.**


## 7. Concluding remarks

The datasets presented in this paper can be used to investigate the spatial and temporal dynamics of

shallow groundwater and to test hydrological models or upscaling approaches. The datasets also allow

comparing groundwater level dynamics and groundwater chemistry. These data can be used in other

studies in the Krycklan catchment to better understand its hydrological functioning or geochemical or

ecological processes. The wells are still in place and accessible, thus, there is the possibility for

continued groundwater level measurements, as well as repeated water sampling. The highly

instrumented sites within the Krycklan catchment provide unique opportunities to study groundwater

dynamics and thepotential findings are relevant for other boreal catchments.





## 8. Data Availability

Dataset 1 and 2 can both be accessed and downloaded via https://www.safedeposit.se/projects/82
(Erdbrügger et al., 2022). The zip-compressed folder (Krycklan_Groundwater_levels_sampling.zip)
includes all the files listed in Table 2

## 9. Author contribution

JE, IvM, JS and KB designed the experiments, JE led the installation of the wells, recording of the water
levels and sampling. JE prepared the manuscript with contributions from all co-authors.

## 10. Competing interests

The authors declare that they have no conflict of interest

## 11. Acknowledgements

We thank everyone involved in the installation and maintenance of the wells, especially Tommy
Andersson, Alexandre Constan, Olivier Dulouard, Corentin Leprince, Johannes Larson, Ola Olafson,
Paul Pister, Viktor Sjöblom, Jacob Smeds, Johannes Tiwari, and Jean-Thomas Wenner. We also thank
SITES and SLU Umea for granting access to the area and research station and providing field equipment,
as well as Ida Manfredson and Charlotta Erefur and the rest of the Svartberget field research station
crew for all the help with logistics and moral support during the fieldwork.

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



## Appendix A: Additional information on the water level measurements

### A.1 Manual water level measurements

We used three different methods to measure the depth between the top of the well and the water level: a bubbler, a water level plopper ("kluk lod"), and an acoustic water level sounder. The so-called "bubbler" consists of a rigid tube connected to flexible tubing. In our case, we used a 1 m long metal tube with a ~ 5mm diameter connected to flexible rubber tubing. The metal tube was inserted into the well and air was blown into the flexible tube. The water level in the well was determined based on the change in the sounds once the metal tube reached the water surface (i.e., as soon as one could hear a "bubbling" noise). This method worked best for shallow groundwater levels since the sound was difficult to discern when the water level was deeper (as well as in noisy circumstances (e.g., strong winds)). Because the water can be blown out of the well when blowing too strongly, leading to a slightly deeper groundwater level, we carefully approached the groundwater surface from above and provided only moderate pressure.

A water-level plopper consists of a small metal cylinder attached to the tip of a measuring tape. It is lowered into a well and produces a "plopping" sound when the cylinder hits the water surface. When the water level was deep (>3 m), the sound was sometimes not audible. Also, for shallow groundwater levels, it often took several tries and "plop" sounds to determine the exact depth to the water level with the measuring tape because it required enough momentum for the cylinder to produce a sufficiently distinguishable sound.

The electronic water level meter or (acoustic) water level sounder emits a sound upon contact with water. In addition, a light switches on. The sensor at the tip of the tape or meter has an open electrical circuit closed when it is in contact with water because of the much lower electric resistivity of water than air. Because the groundwater in the Krycklan area has a relatively low electric conductivity (mean EC from the surveys 48.6 µS/cm), the electronic water level meter sometimes failed and indicated the water surface only after being submerged for about half a meter. The problem was solved after purging the wells or stirring the water in the wells. The low EC in some wells may have originated from the snowmelt water that did not drain or mix much with the other groundwater in the well.

### A.2 Continuous water level measurements

We used Odyssey water level loggers for the continuous water level measurements. These capacitance sensors are based on the difference in the dielectric constants of water and air. The weight at the end of a cable serves as one capacitance plate, and the cable as the second. A change in the area of the second capacitance plate (i.e., the cable in contact with the water) results in a change in the signal. For more information on capacitance sensors, see Larson and Runyan (2009); Dataflow Systems PTY Limited (2012); Guaraglia and Pousa (2014)). We deployed two generations of the same type of loggers, which differ in their dimensions (Figure A 1 (a) and (c)). The new logger bodies (from 2019) are larger and do not fit completely into the well pipes (see Figure A 1 (b) and (Figure A 1 (d)).



The water used for the calibration was taken from a groundwater well to account for local water chemistry (as this has been shown to be a potential error source by Larson and Runyan (2009)) and thus to minimize errors in the calibration. Other error sources identified by Larson and Runyan (2009), like

the potential for a biofilm to build up on cable and counterweight of the loggers, were minimized by cleaning the cables and weights with a cloth during logger read-outs. The building of films varied strongly between wells but did not appear to have an identifiable impact in the data of the water levels and therefore we did not correct for this.

Since the loggers do not show the remaining battery power, it is relatively difficult to predict when they

run out of power. Although a rough estimate says the batteries last for about 18 months, this time can vary considerably with environmental factors and is much shorter during low temperatures. This resulted in incomplete time series for some wells (see Table A 1 in Appendix C).

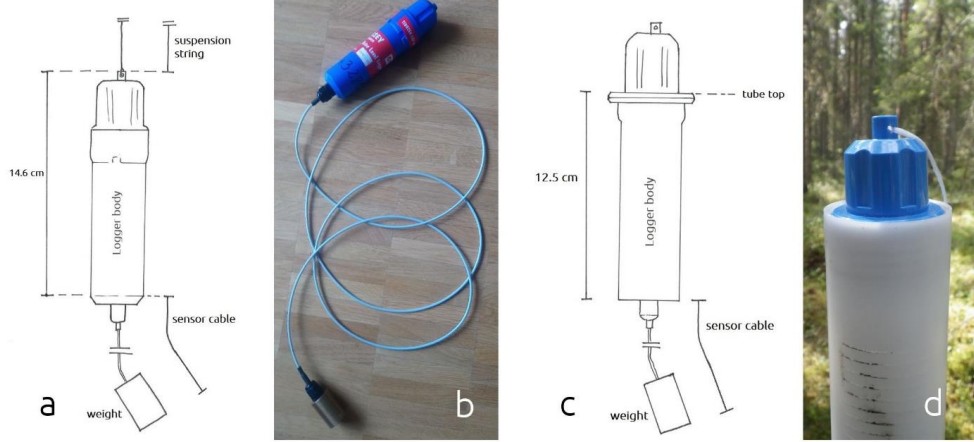

**Figure A 1: Odyssey water level sensors and the respective measures of the logger body and cable lengths for the 2012 Odyssey water level sensor ((a) and (b)) and the 2019 Odyssey water level sensor (c), and the 2019 sensor installed in a well tube (d). Note, drawing is not to scale.**

**Appendix B: Well purging and groundwater sampling**

The well purging and subsequent sampling was done using two peristaltic pumps (see Figure A 2). The volume of water that needed to be pumped from the well during purging was calculated based on the volume of water inside the well, and multiplied by a factor of three to ensure the recharge of fresh groundwater:

$$V_{pump} = 3 * \left( \pi * \left( \frac{d_{well}}{2} \right)^2 \right) * (L_{well} - d_{gw})$$

Where the $V_{pump}$ is the volume to be pumped, $d_{well}$ is the inner well diameter (3.7 cm in our wells), $L_{well}$ is the well depth (from tube top), and $d_{gw}$ is the depth from the well top to the water level.

In some cases, recharge into the well was very slow, and in these cases, we would stop the purging when the well ran dry (i.e., before $V_{pump}$ was reached). All wells (with water in them) were purged twice within the space of two weeks. After the second purging round (31. May 2019) and before the sampling

campaign started (01. July 2019), precipitation was 68.4 mm.

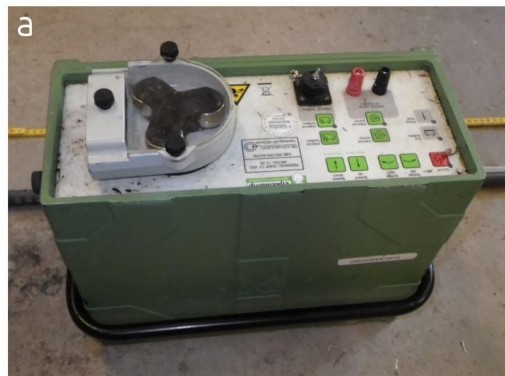
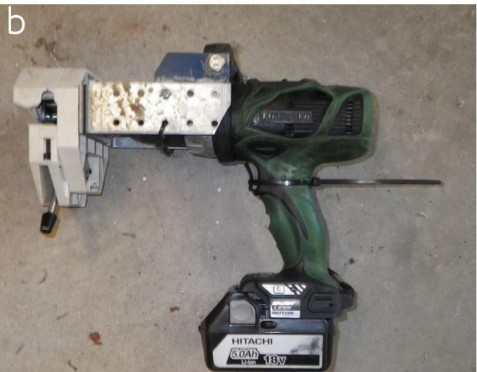

Figure A 2: Photo of peristaltic pump (a) and peristaltic pump mounted on a drill (b) used for purging the wells

For the groundwater sampling, we used a custom-made packer system (Figure A 3) that could seal off

the access to water below and above the part where the sample would be taken. This allowed for sampling at discrete depths. Markers on the tube of the packer system allowed the determination of the sample depth. The packer was designed to sample up to a depth of 6 m. Since the sample opening is

located above the lowest end of the packer, a minimum water level of 40 cm inside the well was necessary to take a sample with the packer system.

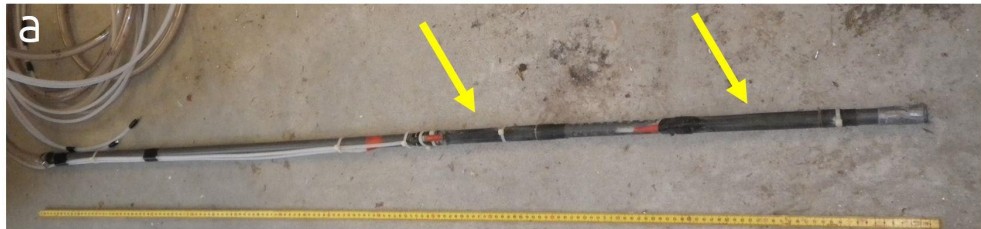

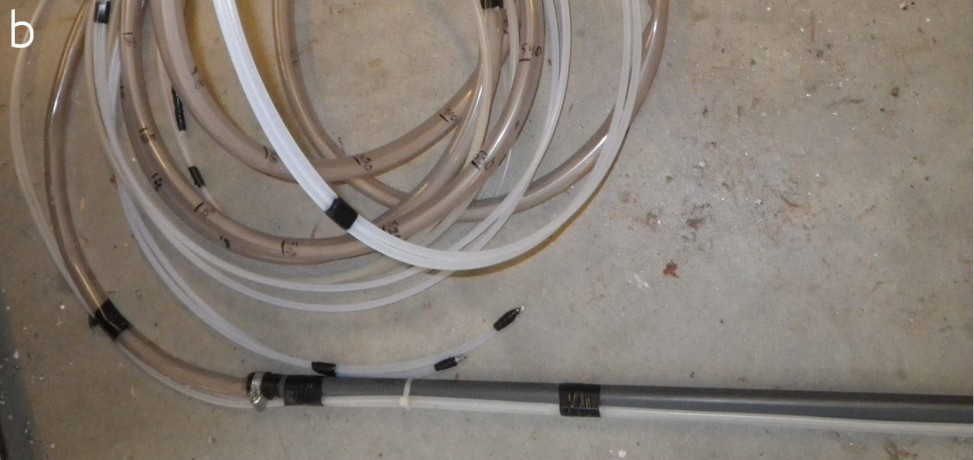


**Figure A 3: Photo of the custom-made straddle packer system with two inflatable parts to isolate specific depths in the wells and to pump water from specific depth (a) and the upper part and tubing system (pumping tube and two air tubes to inflate the flexibles tube parts) of the packer system (b). The yellow arrows in (a) indicate inflatable packer system parts to seal off the water above and below the selected sample depth.**


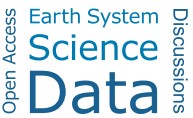

## Appendix C: Logger and manual water level summary

Table A 1: Summary of the recorded water level data per well and the manual measurements for each well. The summary statistics (mean, median, minimum and maximum) were only calculated over the valid data points. Note that for some wells (e.g., A45 and A50) the number of valid data points is very small because the well was dry for most of the study period or the data were flagged as censored data. *non-valid data points and manual outlier include dry well values

| well | Logger data | | level in [m amsl] | | | | Manual measurements | | level in [m amsl] | | | |
|---|---|---|---|---|---|---|---|---|---|---|---|---|
| | Valid data points | Non-valid data points * | mean | median | min | max | Number of valid measurements | Number of outliers * | mean | median | min | max |
| A1 | 0 | 97353 | | | | | 0 | 0 | | | | |
| A2 | 97145 | 258 | 264.18 | 264.16 | 263.61 | 265.07 | 5 | 21 | 264.40 | 264.48 | 263.76 | 264.75 |
| A3 | 90299 | 231 | 261.45 | 261.30 | 260.70 | 262.98 | 16 | 2 | 261.47 | 261.31 | 260.83 | 262.72 |
| A4 | 96838 | 677 | 263.35 | 263.41 | 262.20 | 264.20 | 10 | 15 | 262.99 | 262.88 | 262.59 | 263.77 |
| A5 | 67179 | 11803 | 264.93 | 264.87 | 264.40 | 265.65 | 4 | 13 | 265.09 | 265.08 | 264.88 | 265.33 |
| A6 | 78746 | 234 | 263.47 | 263.43 | 262.04 | 264.29 | 13 | 4 | 263.51 | 263.52 | 263.02 | 263.88 |
| A7 | 71683 | 8653 | 265.85 | 265.79 | 265.64 | 266.45 | 23 | 2 | 265.79 | 265.75 | 265.63 | 266.21 |
| A8 | 72037 | 6947 | 262.10 | 261.94 | 261.50 | 263.21 | 7 | 10 | 262.06 | 262.03 | 261.90 | 262.32 |
| A9 | 97084 | 440 | 261.76 | 261.70 | 260.60 | 263.03 | 16 | 9 | 261.58 | 261.47 | 261.26 | 262.81 |
| A10 | 47317 | 29229 | 262.24 | 262.01 | 261.80 | 263.50 | 3 | 8 | 262.34 | 262.36 | 262.11 | 262.54 |
| A11 | 97040 | 723 | 264.22 | 264.22 | 263.87 | 264.66 | 18 | 5 | 264.16 | 264.18 | 263.90 | 264.45 |
| A12 | 97027 | 486 | 261.20 | 261.19 | 260.39 | 262.32 | 7 | 18 | 260.59 | 260.60 | 260.44 | 260.80 |
| A13 | 70774 | 237 | 265.25 | 265.27 | 264.64 | 265.41 | 19 | 1 | 265.22 | 265.25 | 264.97 | 265.32 |
| A14 | 2953 | 86887 | 264.37 | 264.37 | 264.23 | 264.67 | 2 | 0 | 264.26 | 264.26 | 264.19 | 264.33 |
| A15 | 97055 | 416 | 264.21 | 264.22 | 263.18 | 264.48 | 23 | 3 | 264.23 | 264.24 | 264.08 | 264.32 |
| A16 | 97242 | 187 | 264.11 | 264.14 | 263.51 | 264.34 | 16 | 10 | 264.14 | 264.14 | 263.99 | 264.25 |
| A17 | 57718 | 32496 | 264.22 | 264.17 | 264.00 | 264.65 | 12 | 0 | 264.19 | 264.14 | 263.98 | 264.52 |
| A18 | 72977 | 230 | 261.05 | 260.92 | 260.30 | 262.36 | 1 | 10 | 260.94 | 260.94 | 260.94 | 260.94 |




| well | Logger data | | | | | | Manual measurements | | | | | |
| | Valid data points | Non-valid data points * | level in [m amsl] mean | median | min | max | Number of valid measurements | Number of outliers * | level in [m amsl] mean | median | min | max |
| --- | --- | --- | --- | --- | --- | --- | --- | --- | --- | --- | --- | --- |
| A19 | 78843 | 150 | 261.50 | 261.40 | 260.63 | 262.71 | 10 | 7 | 261.47 | 261.34 | 260.78 | 262.56 |
| A20 | 86289 | 3549 | 263.56 | 263.60 | 263.02 | 263.74 | 15 | 1 | 263.53 | 263.55 | 263.16 | 263.65 |
| A21 | 95942 | 1526 | 261.84 | 261.93 | 260.45 | 262.25 | 16 | 9 | 261.79 | 261.72 | 261.42 | 262.13 |
| A22 | 275 | 78914 | 265.46 | 265.47 | 265.23 | 265.65 | 1 | 0 | 265.22 | 265.22 | 265.22 | 265.22 |
| A23 | 51623 | 230 | 259.39 | 259.35 | 259.02 | 259.96 | 16 | 1 | 259.37 | 259.29 | 259.09 | 259.81 |
| A24 | 0 | 71921 | | | | | 4 | 0 | 265.75 | 265.75 | 265.75 | 265.76 |
| A25 | 240 | 114128 | 264.00 | 263.99 | 263.98 | 264.20 | 2 | 0 | 263.95 | 263.95 | 263.95 | 263.96 |
| A26 | 91847 | 5562 | | | | | 0 | 6 | | | | |
| A27 | 97238 | 271 | 261.03 | 261.08 | 259.44 | 261.63 | 18 | 7 | 260.95 | 260.94 | 260.42 | 261.50 |
| A28 | 79115 | 161 | 257.58 | 257.51 | 257.17 | 258.29 | 16 | 1 | 257.56 | 257.44 | 257.22 | 258.15 |
| A29 | 60092 | 154 | 258.31 | 258.25 | 257.92 | 258.88 | 17 | 0 | 258.29 | 258.21 | 257.98 | 258.75 |
| A30 | 66847 | 150 | 258.13 | 258.16 | 257.57 | 258.50 | 5 | 3 | 258.06 | 258.11 | 257.71 | 258.39 |
| A31 | 75478 | 0 | 263.19 | 263.21 | 262.41 | 263.94 | 9 | 1 | 263.08 | 263.05 | 263.00 | 263.23 |
| A32 | 75405 | 78 | 263.42 | 263.40 | 262.56 | 264.13 | 14 | 1 | 263.36 | 263.34 | 263.18 | 263.61 |
| A33 | 264 | 66577 | 263.31 | 263.28 | 263.08 | 263.61 | 0 | 0 | | | | |
| A34 | 58548 | 1669 | 261.01 | 261.02 | 260.12 | 261.33 | 11 | 6 | 260.98 | 260.96 | 260.92 | 261.15 |
| A35 | 73967 | 5323 | 261.01 | 261.04 | 260.29 | 261.40 | 13 | 4 | 261.03 | 261.01 | 260.95 | 261.15 |
| A36 | 14769 | 16537 | 277.77 | 278.17 | 262.77 | 278.63 | 0 | 0 | | | | |
| A37 | 79058 | 235 | 260.73 | 260.79 | 259.85 | 261.27 | 14 | 3 | 260.76 | 260.78 | 259.98 | 261.01 |
| A38 | 74321 | 153 | 257.16 | 257.15 | 255.57 | 257.35 | 15 | 0 | 257.14 | 257.12 | 256.95 | 257.26 |
| A39 | 0 | 97756 | | | | | 0 | 0 | | | | |
| A40 | 79124 | 160 | 257.01 | 256.92 | 255.91 | 257.50 | 15 | 2 | 257.05 | 256.95 | 256.68 | 257.43 |
| A41 | 30199 | 45368 | 261.40 | 261.21 | 260.76 | 262.86 | 1 | 9 | 260.35 | 260.35 | 260.35 | 260.35 |



| well | Logger data | | | | | | Manual measurements | | | | | |
| | Valid data points | Non-valid data points * | level in [m amsl] | | | | Number of valid measurements | Number of outliers * | level in [m amsl] | | | |
| | | | mean | median | min | max | | | mean | median | min | max |
| --- | --- | --- | --- | --- | --- | --- | --- | --- | --- | --- | --- | --- |
| A42 | 75399 | 74 | 261.52 | 261.38 | 261.15 | 263.12 | 0 | 9 | | | | |
| A43 | 79132 | 154 | 257.12 | 257.06 | 256.20 | 257.85 | 11 | 6 | 257.16 | 257.05 | 256.96 | 257.64 |
| A44 | 65520 | 19891 | 259.35 | 259.04 | 258.56 | 261.67 | 8 | 10 | 259.12 | 258.86 | 258.46 | 260.42 |
| A45 | 16346 | 71051 | 260.87 | 261.01 | 259.99 | 261.29 | 2 | 10 | 260.56 | 260.56 | 259.98 | 261.14 |
| A46 | 79061 | 228 | 257.07 | 257.03 | 256.55 | 257.81 | 11 | 5 | 257.05 | 256.92 | 256.75 | 257.58 |
| A47 | 79068 | 227 | 256.68 | 256.67 | 255.61 | 257.02 | 2 | 15 | 256.66 | 256.66 | 256.59 | 256.74 |
| A48 | 23935 | 99650 | 260.96 | 261.03 | 259.68 | 261.89 | 4 | 9 | 261.18 | 261.22 | 260.53 | 261.76 |
| A49 | 2960 | 94487 | 258.55 | 258.55 | 258.46 | 258.64 | 0 | 0 | | | | |
| A50 | 23010 | 48967 | 261.07 | 260.89 | 260.89 | 262.23 | 0 | 5 | | | | |
| A51 | 32573 | 64715 | 257.09 | 257.08 | 256.38 | 258.04 | 4 | 4 | 256.58 | 256.56 | 256.45 | 256.76 |
| A53 | 68316 | 7243 | 257.08 | 256.76 | 256.42 | 259.28 | 9 | 1 | 257.04 | 256.83 | 256.47 | 258.62 |
| A54 | 56619 | 18909 | 257.37 | 256.98 | 256.58 | 259.23 | 8 | 6 | 257.47 | 257.33 | 256.50 | 258.82 |
| B1 | 75616 | 6502 | 263.88 | 263.81 | 263.04 | 265.20 | 13 | 2 | 263.79 | 263.83 | 263.47 | 264.15 |
| B2 | 66356 | 5438 | 261.87 | 262.07 | 260.92 | 262.30 | 0 | 11 | | | | |
| B3 | 71507 | 286 | 261.53 | 261.43 | 261.01 | 262.43 | 11 | 0 | 261.51 | 261.47 | 261.03 | 261.98 |
| B4 | 75555 | 6570 | 259.39 | 259.27 | 258.65 | 260.56 | 13 | 2 | 259.28 | 259.28 | 258.68 | 260.17 |
| B5 | 65675 | 16455 | 258.45 | 258.28 | 258.02 | 259.66 | 7 | 8 | 258.18 | 258.15 | 257.92 | 258.38 |
| B6 | 56558 | 23909 | 258.77 | 258.59 | 258.41 | 259.95 | 4 | 11 | 258.50 | 258.40 | 258.35 | 258.84 |
| B7 | 63063 | 6419 | 259.25 | 259.14 | 258.44 | 260.44 | 3 | 0 | 258.79 | 258.83 | 258.62 | 258.92 |
| B8 | 68540 | 156 | 259.53 | 259.43 | 258.72 | 260.78 | 9 | 0 | 259.25 | 259.35 | 258.99 | 259.48 |
| B9 | 75628 | 76 | 257.76 | 257.61 | 257.10 | 259.13 | 12 | 3 | 257.70 | 257.61 | 257.13 | 258.58 |
| B10 | 12001 | 63102 | 258.80 | 258.78 | 258.19 | 259.40 | 1 | 8 | 257.57 | 257.57 | 257.57 | 257.57 |
| B11 | 75621 | 6595 | 259.39 | 259.35 | 258.71 | 260.58 | 15 | 0 | 259.33 | 259.35 | 258.89 | 259.72 |


| well | Logger data | | level in [m amsl] | | | | Manual measurements | | level in [m amsl] | | | |
|------|------|------|------|------|------|------|------|------|------|------|------|------|
| | Valid data points | Non-valid data points * | mean | median | min | max | Number of valid measurements | Number of outliers * | mean | median | min | max |
| B12 | 74599 | 7516 | 257.51 | 257.47 | 257.00 | 258.45 | 5 | 10 | 257.32 | 257.47 | 257.00 | 257.62 |
| B13 | 75622 | 6496 | 257.26 | 257.25 | 256.78 | 258.35 | 4 | 11 | 257.34 | 257.34 | 257.18 | 257.49 |
| B14 | 68391 | 7308 | 257.24 | 257.25 | 255.94 | 257.54 | 0 | 15 | | | | |
| B15 | 23385 | 52314 | 256.40 | 256.26 | 255.97 | 257.28 | 0 | 15 | | | | |
| B16 | 75618 | 80 | 256.85 | 256.83 | 255.83 | 257.30 | 10 | 5 | 256.70 | 256.78 | 256.30 | 256.98 |
| B17 | 67719 | 7982 | 255.54 | 255.38 | 255.14 | 256.68 | 8 | 7 | 255.45 | 255.37 | 255.18 | 255.90 |
| B18 | 75624 | 78 | 254.71 | 254.67 | 254.11 | 255.51 | 14 | 1 | 254.59 | 254.62 | 254.13 | 255.06 |
| B19 | 69087 | 6614 | 253.68 | 253.53 | 253.23 | 254.28 | 11 | 4 | 253.60 | 253.49 | 253.15 | 254.18 |
| B20 | 40737 | 816 | 253.53 | 253.41 | 252.85 | 254.50 | 13 | 0 | 253.36 | 253.37 | 252.82 | 254.02 |