# Peer review of "Shallow groundwater level time series and a groundwater chemistry survey from a boreal headwater catchment, Krycklan, Sweden"

_Earth System Science Data, 2022_

## Author Comment (AC1)

**Reviewer I**

Erdbrügger et al. present a database of groundwater level recorded at 75 wells in a Swedish experimental catchment for two years, from July 2018 to November 2020. They also present the hydrochemistry acquired in the wells during one sampling campaign in summer 2019. A full description from the setup of the well network to the data quality are presented. Additionally, some results illustrate the interest of having such hydrological/hydrogeological data published.

As it is more and more important having access to such information, the publication of these data is for me very relevant. However, I also think that the manuscript, as submitted, needs a substantial improvement before being published in ESSD. Please see below my general comments.

*We thank the reviewer for seeing the value of these types of datasets. We respond to the comments in red font below.*

A clear definition of what is a "shallow groundwater" should be given. Indeed, by reading the entire manuscript we are lost in between "shallow" GW, GW or even deep GW. You should more clearly explain what are the different groundwater that you had access with your wells in this catchment. If you only look at shallow GW, please explain it more clearly.

*Thank you for this important comment.*

*We define shallow groundwater as unconfined groundwater in the soil or regolith, or perched above a less permeable layer. It flows faster, is younger, and more important for streamflow generation during events than deeper groundwater (e.g., from the bedrock or deeper layers).*

*More specifically, for our study site and thus this manuscript, we refer to shallow groundwater as groundwater that it is located within the glacial till. In other words, what we mean by shallow groundwater is that the water table is within ten meter of the soil surface. This shallow groundwater feeds a network of headwater streams, and some fens. Our groundwater tables were as deep as 6 m below the ground surface. Indeed there is also deeper groundwater in the bedrock in Krycklan that is much older (Kolbe et al., 2020) and is important for baseflow, particularly in the larger streams.*

*We realize that in the current manuscript, we referred to shallow and deeper groundwater in relation to both the range of well depths (or type of aquifer) and the depth of the groundwater samples and understand that this is confusing. All of the samples were taken from what we consider shallow groundwater. We will revise the text and use different wording to indicate the samples taken from the uppermost part of the well (i.e., upper most part of the shallow groundwater) and the lower part of the well (i.e., the lower part of the shallow groundwater).*

The structure of the regolith (soil-saprolite-fractured bedrock-fresh bedrock) of the catchment needs to be presented with a more rigorous and complete description. More specifically, the very short description of the soil is not clear at all and do not give the minimum information we need to link with the GW dynamics or with the water chemistry. You should provide information on the spatial variability

of the soil properties (depth, WRB soil type, some basic pedological parameters and if available physical parameters related to hydrology) at both sub catchment and hillslope scales.

*This was indeed an oversight. We will provide a description of the podzolic soils that have developed on the glacial tills and the underlying bedrock. Furthermore, references to more complete descriptions will be provided.*

*In short, the landscape is strongly influence by the last glaciation, which left glacial tills up to ten meters tick over the metamorphic bedrock. Podzols developed in this glacial till; at the base of the slopes, organogenic soils developed.*

   The GW chemistry was only recorded during one sampling campaign in July 2019 which is not representative of the complete GW level range you monitored for 2 years. You should explain what you did expect from this sampling and what is the added value having these data published together with the GW level. The different wells were sampled at different dates and during this period precipitation happened (about 28mm, which is not negligible, isn't it?): how different were the hydrological conditions during these sampling dates? How could such differences affect the spatial variability you observed and the GW connectivity in between wells?

*We recognize that the groundwater chemistry data are only from one sampling occasion, and that the samples took several weeks. The groundwater sampling required a lot of time, equipment and manpower and therefore could not be completed in one day, or repeated multiple times during the study period. These data nevertheless gave a good impression of the general chemistry in the groundwater during baseflow conditions and the spatial variability in shallow groundwater chemistry. We think that this information is useful and feel it is of value to present these data as a complement to the more comprehensive water level information because:*

1) *there is a general lack of information about the chemistry of shallow groundwater across small catchments (see Kiewiet et al. (2019) but also Penna and van Meerveld (2019))*
2) *it complements the existing long term monitoring of soil water on the S-transect and the stream chemistry (Laudon et al., 2013). These studies provide more information on the temporal variation in chemistry but less detail on the spatial variation.*
3) *this data can serve as a baseline for future sampling campaigns or to determine the most important wells to sample in future campaigns. In other words, it is a starting point, and with these data being available, it will be possible for others to extend these data to obtain a more temporally complete picture of the groundwater chemistry.*
4) *this information can be useful for people who want to use the groundwater dataset in a groundwater or catchment model for the Krycklan catchment or use this data for virtual experiments,*

*We can make these points clearer in the revised version of the paper.*

*It would have been nice to complete the sampling within one day but this was simply not possible. As for the 28 mm of rain falling during the month of July when the samples were collected, we do not think that the addition of weakly buffered rainfall infiltrating through in many places more than a meter of soil will significantly change the chemistry of the groundwater. Furthermore, Kiewiet et al. (2019) showed that the chemistry of shallow groundwater in a Swiss headwater catchment did not vary much during the summer-fall and that the spatial variation in shallow groundwater chemistry was much larger than the*

*temporal variation. But of course, some variation is to be expected. That is why we carefully describe the conditions during the sampling period.*

The size of the manuscript should be reduced by removing most of the tables. Indeed, the table information is always described in the text (redundancy). Moreover, the information that is presented in the tables could be more relevant in direct link to (or inside) the files provided online under the "safedeposit" website. Some figures could be merged to reduce its number (see below).

*This is perhaps a matter of taste. We preferred to have tables in addition to the text because they provide a compact and structured overview of all the information. But we agree that the length of the manuscript can be reduced. We aim to do this by moving some of the marked tables to the appendix rather than removing them entirely. As for the figures, we will merge them as suggested below.*

Online files need to be improved (information missing, not clear enough, language harmonization)

*Thanks for making us aware of these issues. We will provide more comprehensive information, describe what is there more clearly and ensure that all the information is in English.*

Please find below some more detailed suggestions/comments:

….Title

The name of the catchment and the country should appear in the title

*We will add the information*

Introduction

Lines 28-29: I don't see the choice of N and Hg relevant when speaking about GW solutes. You should find a better choice.

*While N and Hg may not be classic groundwater solutes, both have been extensively studied in the Krycklan catchment. For this reason, we included these constituents in the measurement campaign. We will adjust the formulation to reflect that these examples refer mainly to boreal ecosystems.*

Line 43: "…understanding of hydrological…"

*We will correct this.*

Line 97: "Shallow" GW?

*We will provide a more explicit definition of "shallow groundwater" (see also our longer reply above)*

description of the study area

Line 107: catchment area?

*We will add the catchment area (6790 ha).*

Line 108: "long-term data": give the initial and final dates that cover the time series

*In the revised version, we will provide information on when the measurements began. We will also refer more explicitly to publications where detailed information on when specific measurements started can be found* (e.g., Laudon et al., 2013, 2021).

Line 125-126: not clear at all

*Thank you for pointing this out. We will rewrite this part of the study area description.*

Line 127: 6m depth, is this soil developed on deposited material (colluvium, alluvium…)?

*All groundwater wells for which the data are reported in this paper were located in the till overburden. Some wells drilled into the bedrock exist in the catchment* (Kolbe et al., 2020; Laudon et al., 2013)*, but these are not in the areas covered by our detailed well network. We will mention this specifically in the text.*

*We will include a more extensive description of the soils and bedrock in the area of the well network.*

Line 137: The ICOS station should be presented on the map in figure 1

*We will add the position on the map in Figure 1*

Groundwater wells

Lines 176-177: better to give the range than the average

*We will add the ranges*

Dataset 1

Lines 341-243: why not using the same procedure for all wells?

*We used the bubbler when the water table was close to the surface because we found this to be the most accurate method when the water table was close to the surface. Especially for the very low EC conditions, the acoustic water level sounder did not always provide a clear sound and the measurements had to be repeated several times. When the water level was deeper, the signal of the bubbler was sometimes too weak so that for these measurements the acoustic sounder or plopper provided better data. We will clarify this in the revised version of the manuscript.*

Lines 266-270: precision of the measure by the automatic sensors?

*The resolution as given by Dataflow Systems Ltd* (2021) *is 0.8 mm*

*We will add this information to the logger section.*

Line 302: The first step for the manual selection should be shown in figure 5 to clarify all the used procedure.

*We will add an extra panel to figure 5 to show all the data, the original trendline, and which data points were excluded in this first step.*

Lines 306-309: this is not clear to me. Please explain why this can happen. Is it because this measurement is not always as sensitive even if correctly done?

*We now see that this sentence was unclear. We meant that the intercept/correction was not calculated if there were fewer than two valid data points. We will revise the text*

Line 342: "recovery time", should be interesting to know the necessary time to recover at each well to show the spatial heterogeneity of some hydraulic properties. This could be one of the example results, for instance.

*Yes, this could be possible, but it is not something that we have done so far. As so many things could be done with these data, we prefer to stick with the examples we have presented so far (also to not make the paper even longer), and prefer to leave it to others to use the data and these types of analyses in their studies.*

   Dataset 2

Line 393: how often, the wells were dried and in what hydrological state?

*The wells were pumped dry two times after the snowmelt peak in the beginning of May. We will adjust the formulation so that this is clearer.*

Lines 397-399: the purging description (lines 390-395) should appear in this paragraph because it is a part of the sampling protocol.

*We will integrate the purging process description into the sampling protocol.*

Line 407: which should correspond to the shallower part oof the GW, shouldn't it?

*Yes, this would refer to the uppermost portion of the groundwater at the respective location.*

Line 412: what pumping speed? Was it low enough to completely avoid this effect? How did you evaluated this for all wells and how variable was it for all wells?

*The pumping speed was adjusted manually but the speed was not noted for the individual wells. As mentioned in the next sentence, aeration could not always be avoided. This effect was noted in the sample protocol giving a qualitative estimation of sample quality. We will add the information on sample quality indication and clarify the formulation.*

   Example results

Line 432: is it not mainly transpiration that would affect GW level? Can evaporation from the surface of the soil impact the GW level?

*It is indeed mostly transpiration that affects the groundwater level. Nevertheless, as this area is very moist and groundwater levels are very close to the surface, a small effect of evaporation cannot be excluded. Thus, we like to use the more general term evapotranspiration. In a more detailed study, it would probably be possible to quantify the effects of each mechanism on the groundwater level variation.*

Lines 433-434: how many wells and why these ones?

*This effect was seen for almost all wells at one time or another but the effect was of course much clearer for some of them. The effect is likely due to the transpiration but could early in the year also be caused by freeze and thaw effects of the surface snowmelt water, which then infiltrates. We have not analyzed these patterns in detail and leave the detailed analysis for a later study.*

*Temperature effects can also change the viscosity and lateral flow to the stream (e.g., Schwab et al., 2016).*

Line 444: The deep GW was not defined previously

*We were referring to the lower groundwater sample (at the locations where two samples were taken). We will adjust the formulation. See also longer discussion about shallow and deep groundwater above.*

Line 445: what statistical test did you used to estimate the significance?

*We apologize, but we did not do statistical tests and only compared values. The formulation will be changed accordingly*

Line 446: is it not 12.5 because in the figure the range is closer to 10. If not 1.25 is in the same order of magnitude that the mean analytical error we have with standard isotope analyzer, then not really large.

*Thank you for finding this typo. It is correct that it should be 12.5. We will change this in the text.*

Tables

Tables 1 and 2 are not necessary

*We propose to keep Table 1 and 2 but to move them to the appendix as suggested by Reviewer II.*

Tables 4 and 5 should be removed and their information added to the related online files

*We propose to move the tables to the Appendix, in addition to having the information included in the online files.*

Table 6 is not necessary as fully described in the text at 4.4. The caption is not detailed enough. Is it for manual or logger data?

*The table refers to the classification of logger data points. We will adapt the caption accordingly. As explained in our response above, we will remove the table, along with others, from the body of the manuscript and place them in the appendix*

Table 8 not needed

*We will move this table to the appendix. As explained above, we find it useful to have these overview tables so that the reader doesn't have to go through the text to find the information. However, we agree that it is not very necessary to have this as part of the text.*

Figures

Figure 1 and 2 should be merged and well labelling added on Figure 2

*We feel that merging Figures 1 and 2 would lead to an overload of information in one consolidated figure. We, therefore, prefer (also for easier placement of the figures) to keep both figures separate.*

*We will add labels to the wells in Figure 2*

Figure 4 is not clear. All the information provided in the figure caption should be indicated on the figure too.

*We will add labels in Figure 4.*

Figure 6 and 7 should be merged to show the 6 different classes together.

*Since Figure 6 and 7 show very different time intervals, we feel that they are rather difficult to combine in a way that still shows the result that we want to highlight. Figure 7 shows an effect that only concerns five wells, while Figure 6 applies to all wells, which is why we feel they are better kept separate.*

The legend should be added on figure 8

*We will add the legend.*

Appendix A should be put in the online repository with the other files.

*We consider this information to be more directly related to the information related in the descriptions and would therefore prefer to keep Appendix A.*

Online files

Kryckland_gw_levels.csv: avoid the acronyms and put together the column for mnl

*The acronyms were necessary to conform to requirements imposed by the use of the Shape-file format (especially the limitation of column name length) for geospatial analyses (a commonly used format which can be handled by most GIS programs). Though not strictly necessary for the csv data, we decided to use the acronyms for an easier reintegration of the data in a GIS program and to avoid the renaming that is needed to use and save data in the Shape-format.*

Kryckland_gw_sampling.csv and Kryckland_gw_chemistry.csv should be merged in one file

*Due to differences in the format of the files, we prefer to keep these as separate files. They are, however, linked via the well names.*

Field_protocol.csv is not clear because some column (like Y and Z) do not have title and what means g/d in column N?

*We will provide a more detailed description of the column contents.*

*In short column, X and Y referred to the X- and Y-coordinate, column Z to elevation and g/d to the perceived quality of the sample (g-Good, d- doubtful, b- bad). We will rename the latter to "quality".*

Lab_analysis_description.pdf: harmonized the language to English

*We will provide a translation for the German descriptions*

*References:*

Dataflow Systems Ltd: Odyssey Capacitance Water Level Logger, [online] Available from: http://odysseydatarecording.com/index.php?route=product/product&path=59&product_id=50, 2021.

Kiewiet, L., Freyberg, J. and Meerveld, H. J. (Ilja): Spatiotemporal variability in hydrochemistry of shallow groundwater in a small pre-alpine catchment: The importance of landscape elements, Hydrol. Process., 33(19), 2502–2522, doi:10.1002/hyp.13517, 2019.

Kolbe, T., Marçais, J., de Dreuzy, J. R., Labasque, T. and Bishop, K.: Lagged rejuvenation of groundwater indicates internal flow structures and hydrological connectivity, Hydrol. Process., 34(10), 2176–2189, doi:10.1002/hyp.13753, 2020.

Laudon, H., Taberman, I., Ågren, A., Futter, M., Ottosson-Löfvenius, M. and Bishop, K.: The Krycklan Catchment Study - A flagship infrastructure for hydrology, biogeochemistry, and climate research in the boreal landscape, Water Resour. Res., 49(10), 7154–7158, doi:10.1002/wrcr.20520, 2013.

Laudon, H., Hasselquist, E. M., Peichl, M., Lindgren, K., Sponseller, R., Lidman, F., Kuglerová, L., Hasselquist, N. J., Bishop, K., Nilsson, M. B. and Ågren, A. M.: Northern landscapes in transition: Evidence, approach and ways forward using the Krycklan Catchment Study, Hydrol. Process., 35(4), 1–15, doi:10.1002/hyp.14170, 2021.

Penna, D. and van Meerveld, H. J.: Spatial variability in the isotopic composition of water in small catchments and its effect on hydrograph separation, Wiley Interdiscip. Rev. Water, 6(5), 1–33, doi:10.1002/wat2.1367, 2019.

Schwab, M., Klaus, J., Pfister, L. and Weiler, M.: Diel discharge cycles explained through viscosity fluctuations in riparian inflow, Water Resour. Res., (52), 8744–8755, doi:10.1002/2016WR018626, 2016.

---

## Author Comment (AC2)

**Reviewer II**

Summary and general comments:

In this manuscript, Erdbrügger et al. installed a monitoring network of groundwater wells in the Krycklan catchment in Northern Sweden and produced two datasets which combined hydrometric and hydrochemical data: groundwater levels and groundwater chemistry. The datasets can be a useful reference to test models that simulate groundwater dynamics and to evaluate subsurface hydrological connectivity. The manuscript is generally well-written and addresses two important datasets that is suitable for the Earth System Science Data. Thus, I can recommend a acceptance after a minor revision.

*We thank the reviewer for these positive words and the useful comment to which we respond in red font below.*

Line-to-line comments:

Page 1, line 9-10, Abstract, add a sentence to summarize the shortcomings of the existing research.

*We will add such a sentence in the revised version of the manuscript*

Page 1, line 10-21, Abstract, the methodology could be streamlined appropriately.

*We will revise the abstract and streamline the information.*

Tables, in general: tables in the manuscript should be spaced appropriately to fit the content; It is recommended to use three-line tables.

*We will reformat the tables to make the text more readable.*

Page 4, Table1, references can be placed in the last column.

*We will move the references to the last column*

Page 4 and 5, Table1 and Table 2 can be used as an appendix.

*We will move Table 1 and 2 to the Appendix*

Page 5, line 107, label the location of the Umea in Figure 1 or delete 'about 60 km inland from Umea'.

*We will add the approximate coordinates of the Krycklan catchment (64°140N, 19°460E) and will label the location of Umeå in Figure 1.*

Page 7, Figure 1 is missing the graticule.

*We omitted the graticule in Figure 1 to avoid clutter and information overload in the figure. We will add the coordinates of the Krycklan catchment and provide the information on the map coordinate system. As all groundwater well coordinates are provided in the dataset and the Krycklan catchment data are openly available for download and further detailed analysis, we prefer to not include this information in this figure to keep it easier to "read".*

Figures in this manuscript (Figures 1, Figure 2, Figure 10, and Figure11), in general: reduce the size of the scale bar and north arrow to make the Figures more coordinated.

*We will adjust the sizes of the scale bar and North arrow in the figures.*

Page 17 and 19, Figure 6 and Figure 7, the labels in the Figures are too small to read.

*We will increase the font size of the labels in both figures*

---

## Author Response (AR1)

**Reviewer I**

Erdbrügger et al. present a database of groundwater level recorded at 75 wells in a Swedish experimental catchment for two years, from July 2018 to November 2020. They also present the hydrochemistry acquired in the wells during one sampling campaign in summer 2019. A full description from the setup of the well network to the data quality are presented. Additionally, some results illustrate the interest of having such hydrological/hydrogeological data published.

As it is more and more important having access to such information, the publication of these data is for me very relevant. However, I also think that the manuscript, as submitted, needs a substantial improvement before being published in ESSD. Please see below my general comments.

*We thank the reviewer for seeing the value of these types of datasets. We respond to the comments in red font below.*

A clear definition of what is a "shallow groundwater" should be given. Indeed, by reading the entire manuscript we are lost in between "shallow" GW, GW or even deep GW. You should more clearly explain what are the different groundwater that you had access with your wells in this catchment. If you only look at shallow GW, please explain it more clearly.

*Thank you for this important comment.*

*We define shallow groundwater as unconfined groundwater in the soil or regolith, or perched above a less permeable layer. It flows faster, is younger, and is more important for streamflow generation during events than deeper groundwater (e.g., from the bedrock or deeper layers).*

*More specifically, for our study site, and thus this manuscript, we refer to shallow groundwater as groundwater located within the glacial till. In other words, what we mean by shallow groundwater is an unconfined aquifer with a water table within ten meter of the soil surface. This shallow groundwater feeds a network of headwater streams and some fens. Our groundwater tables were as deep as 6 m below the ground surface. Indeed there is also deeper groundwater in the bedrock in Krycklan that is much older* (Kolbe et al., 2020) *and is important for baseflow, particularly in the larger streams.*

*In the first manuscript, we referred to shallow and deeper groundwater in relation to both the range of well depths (or type of aquifer) and the depth of the groundwater samples and understand that this was confusing. All of the samples were taken from what we consider shallow groundwater. We revised the text and now use different wording to indicate the samples taken from the uppermost part of the well (i.e., upper most part of the shallow groundwater) and the lower part of the well (i.e., the lower part of the shallow groundwater), see lines 32, 127-129, 402-408 of the revised text.*

The structure of the regolith (soil-saprolite-fractured bedrock-fresh bedrock) of the catchment needs to be presented with a more rigorous and complete description. More specifically, the very short description of the soil is not clear at all and do not give the minimum information we need to link with the GW dynamics or with the water chemistry. You should provide information on the spatial variability

of the soil properties (depth, WRB soil type, some basic pedological parameters and if available physical parameters related to hydrology) at both sub catchment and hillslope scales.

*This was indeed an oversight. We now provide a description of the podzolic soils that have developed on the glacial tills and the underlying bedrock and references to more complete descriptions (lines 123-133).*

*In short, the landscape is strongly influenced by the last glaciation, which left glacial tills up to ten meters thick over the metamorphic bedrock. Podzols developed in this glacial till; at the base of the slopes, organogenic soils developed.*

The GW chemistry was only recorded during one sampling campaign in July 2019 which is not representative of the complete GW level range you monitored for 2 years. You should explain what you did expect from this sampling and what is the added value having these data published together with the GW level. The different wells were sampled at different dates and during this period precipitation happened (about 28mm, which is not negligible, isn't it?): how different were the hydrological conditions during these sampling dates? How could such differences affect the spatial variability you observed and the GW connectivity in between wells?

*We recognize that the groundwater chemistry data are only from one sampling occasion, and that the sampling campaign took several weeks. The groundwater sampling required a lot of time, equipment and workforce and, therefore, could not be completed in one day, or repeated multiple times during the study period. These data, nevertheless, gave a good impression of the general chemistry in the groundwater during baseflow conditions and the spatial variability in shallow groundwater chemistry. We think that this information is useful and feel that it is of value to present these data as a complement to the more comprehensive water level information because …*

1) *… there is a general lack of information about the spatial variation in the chemistry of shallow groundwater across small catchments (see Kiewiet et al. (2019) but also Penna and van Meerveld (2019))*

2) *… it complements the existing long-term monitoring of soil water on the S-transect and stream chemistry (Laudon et al., 2013). Those S-transect studies provide more information on the temporal variation in chemistry, but less detail on the spatial variation across the landscape, or the situation below the zone of transient saturation (the upper meter of the soil which has been collected routinely along the S-transect for over a decade).*

3) *… the chemistry data in our study can serve as a baseline for future sampling campaigns or to determine the most important wells to sample in future campaigns. In other words, it is a starting point, and with these data being available, it will be possible for others to extend the dataset to obtain a more temporally complete picture of the groundwater chemistry.*

4) *… this information can be useful for people who want to use the groundwater dataset in a groundwater or catchment model for the Krycklan catchment or use this data for virtual experiments,*

*We clarify these points in the revised version of the paper (lines 465-472).*

*It would have been nice to complete the sampling within one day, but this was simply not possible. As for the 28 mm of rain falling during the month of July when the samples were collected, we do not think that the addition of weakly buffered rainfall infiltrating through more than a meter of soil will significantly change the chemistry of the groundwater at the depths we have sampled. Furthermore, Kiewiet et al.*

*(2019) showed that the chemistry of shallow groundwater in a Swiss headwater catchment did not vary much during the summer-fall sampling period and that the spatial variation in shallow groundwater chemistry was much larger than the temporal variation. But, of course, some variation is to be expected. That is why we carefully describe the conditions during the sampling period.*

The size of the manuscript should be reduced by removing most of the tables. Indeed, the table information is always described in the text (redundancy). Moreover, the information that is presented in the tables could be more relevant in direct link to (or inside) the files provided online under the "safedeposit" website. Some figures could be merged to reduce its number (see below).

*We would argue that this might be a matter of taste. We preferred tables in addition to the text because they provide a compact and structured overview of all the information. However, we agree that the length of the manuscript can be reduced. We did this by moving some of the marked tables to the appendix rather than removing them entirely. As for the figures, please see the detailed answers below.*

Online files need to be improved (information missing, not clear enough, language harmonization)

*Thanks for making us aware of these issues. We provided more comprehensive information, describing what is there more clearly and ensured that all the information is in English. The files have been submitted to the repository to be updated.*

Please find below some more detailed suggestions/comments:

….Title

The name of the catchment and the country should appear in the title

*We added the information - Line 1-3*

Introduction

Lines 28-29: I don't see the choice of N and Hg relevant when speaking about GW solutes. You should find a better choice.

*While N and Hg may not be classic groundwater solutes, both have been extensively studied in the Krycklan catchment. For this reason, we included these constituents in the measurement campaign. We did adjust the formulation to reflect that these examples refer mainly to boreal ecosystems (Lines 32-34)*

Line 43: "…understanding of hydrological…"

*We corrected this (Line 50)*

Line 97: "Shallow" GW?

*We provided a more explicit definition of "shallow groundwater" (see also our longer reply above; Line 31-36)*

description of the study area

Line 107: catchment area?

*We added the catchment area (6790 ha; Line 114)*

Line 108: "long-term data": give the initial and final dates that cover the time series

*In the revised version, we provide information on when the measurements began. We also refer more explicitly to publications where detailed information on when specific measurements started can be found* (e.g., Laudon et al., 2013, 2021).

*Lines 119, 141-147*

Line 125-126: not clear at all

*Thank you for pointing this out. We rewrote this part of the study area description (Lines 143-149)*

Line 127: 6m depth, is this soil developed on deposited material (colluvium, alluvium…)?

*All groundwater wells for which the data are reported in this paper were located in the till overburden. Some wells drilled into the bedrock exist in the catchment* (Kolbe et al., 2020; Laudon et al., 2013)*, but these are not in the areas covered by our detailed well network. We now mention this specifically in the text.*

*We included a more extensive description of the soils and bedrock in the area of the well network.*

*See Lines 124-127, 130-133*

Line 137: The ICOS station should be presented on the map in figure 1

*We added the position on the map in Figure 1.*

Groundwater wells

Lines 176-177: better to give the range than the average

*We added the ranges (Line 186-189)*

Dataset 1

Lines 341-243: why not using the same procedure for all wells?

*We used the bubbler when the water table was close to the surface because we found this to be the most accurate method when the water table was close to the surface. Especially for the very low EC conditions, the acoustic water level sounder did not always provide a clear sound and the measurements had to be repeated several times (see also Appendix A). When the water level was deeper, the signal of the bubbler was sometimes too weak so that for these measurements the acoustic sounder or plopper provided better data. We clarified this in the revised version of the manuscript (Lines 252-257).*

Lines 266-270: precision of the measure by the automatic sensors?

*The resolution, as given by Dataflow Systems Ltd* (2021)*, is 0.8 mm*

*We added this information to the logger section (Line 267)*

Line 302: The first step for the manual selection should be shown in figure 5 to clarify all the used procedure.

*We changed one example in figure 5 to show all the data, the uncorrected data points, and which data points were excluded in this first step. In other words, we now show which datapoints were excluded in the first step.*

Lines 306-309: this is not clear to me. Please explain why this can happen. Is it because this measurement is not always as sensitive even if correctly done?

*We now saw that this sentence was unclear. We meant that the intercept/correction was not calculated if there were fewer than two valid data points. We revised the text (Line 314-319).*

Line 342: "recovery time", should be interesting to know the necessary time to recover at each well to show the spatial heterogeneity of some hydraulic properties. This could be one of the example results, for instance.

*Yes, this could be interesting, but it is not something that we have done so far. The data allows for a lot of further analyses, but here we prefer to stick with the examples we have presented so far (also not to make the paper even longer), and prefer to leave it to others to use the data for these types of analyses in their studies.*

Dataset 2

Line 393: how often, the wells were dried and in what hydrological state?

*The wells were pumped dry two times after the snowmelt peak in the beginning of May. We adjusted the formulation so that this is clearer (Lines 399, 401)*

Lines 397-399: the purging description (lines 390-395) should appear in this paragraph because it is a part of the sampling protocol.

*We integrated the purging process description into the sampling protocol (Lines 399-404)*

Line 407: which should correspond to the shallower part oof the GW, shouldn't it?

*Yes, this would refer to the uppermost portion of the groundwater at the respective location. We clarified this in the text (Line 408)*

Line 412: what pumping speed? Was it low enough to completely avoid this effect? How did you evaluated this for all wells and how variable was it for all wells?

*The pumping speed was adjusted manually but the speed was not recorded for the individual wells. As mentioned in the next sentence, aeration could not always be avoided. This effect was noted in the sample protocol giving a qualitative estimation of sample quality. We added the information on sample quality indication and clarified the formulation (see Lines 412-417).*

Example results

Line 432: is it not mainly transpiration that would affect GW level? Can evaporation from the surface of the soil impact the GW level?

*It is indeed mostly transpiration that affects the groundwater level. Nevertheless, as this area is very moist and groundwater levels are very close to the surface, a small effect of evaporation cannot be excluded. Thus, we prefer to use the more general term evapotranspiration. In a more detailed study, it would probably be possible to quantify the effects of each mechanism on the groundwater level variation.*

Lines 433-434: how many wells and why these ones?

*This effect was seen for almost all wells at one time or another but it was of course much clearer for some of them. The effect is likely due to the transpiration but could early in the year also be caused by diurnal cycles of surface snowmelt, which then infiltrates. Temperature effects can also change the viscosity and lateral flow to the stream (e.g., Schwab et al., 2016). We have not analyzed these patterns in detail and leave the detailed analysis for a later study. We now clarify in the text that we see these variations for most wells (Line 437-440).*

Line 444: The deep GW was not defined previously

*We were referring to the lower shallow groundwater sample (at the locations where two samples were taken). We adjusted the formulation. See also longer discussion about shallow and deep groundwater above (Line 450).*

Line 445: what statistical test did you used to estimate the significance?

*We apologize, but we only compared values and did not do statistical tests. The formulation was changed accordingly (Line 490-491).*

Line 446: is it not 12.5 because in the figure the range is closer to 10. If not 1.25 is in the same order of magnitude that the mean analytical error we have with standard isotope analyzer, then not really large.

*Thank you for spotting this typo. It is correct that it should be 12.5. We changed this in the text (Line 452).*

Tables

Tables 1 and 2 are not necessary

*We kept Table 1 and 2 but moved them to the appendix as suggested by Reviewer II.*

Tables 4 and 5 should be removed and their information added to the related online files

*We moved the tables to the Appendix, in addition to having the information included in the online files.*

Table 6 is not necessary as fully described in the text at 4.4. The caption is not detailed enough. Is it for manual or logger data?

*The table refers to the classification of logger data points. We adapted the caption accordingly. As explained in our response above, we removed the table, along with others, from the body of the manuscript and placed them in the appendix*

Table 8 not needed

*We moved this table to the appendix. As explained above, we find it useful to have these overview tables so that the reader doesn't have to go through the text to find the information. However, we agree that having this as part of the main text is unnecessary.*

Figures

Figure 1 and 2 should be merged and well labelling added on Figure 2

*We tried merging Figures 1 and 2 but it led to an overload of information in one figure. We, therefore, (also for easier placement of the figures) kept both figures separate.*

*We added labels to the wells in Figure 2*

Figure 4 is not clear. All the information provided in the figure caption should be indicated on the figure too.

*We added labels in Figure 4.*

Figure 6 and 7 should be merged to show the 6 different classes together.

*Since Figure 6 and 7 show very different time intervals, we feel that they are rather difficult to combine in a way that still shows the result that we want to highlight. Figure 7 shows an effect that only concerns five wells, while Figure 6 applies to all wells, which is why we kept them separate.*

The legend should be added on figure 8

*We added the legend.*

Appendix A should be put in the online repository with the other files.

*We consider this information to be more directly related to the information related in the descriptions and therefore kept Appendix A (now Appendix E).*

Online files

Kryckland_gw_levels.csv: avoid the acronyms and put together the column for mnl

*The acronyms were necessary to conform to requirements imposed by the use of the Shape-file format (especially the limitation of column name length) for geospatial analyses (a commonly used format which can be handled by most GIS programs). Though not strictly necessary for the csv data, we decided to use the acronyms for an easier reintegration of the data in a GIS program and to avoid the renaming that is needed to use and save data in the Shape-format.*

Kryckland_gw_sampling.csv and Kryckland_gw_chemistry.csv should be merged in one file

*Due to differences in the format of the files, we prefer to keep these as separate files. They are, however, linked via the well names.*

Field_protocol.csv is not clear because some column (like Y and Z) do not have title and what means g/d in column N?

*We provided a more detailed description of the column contents.*

*In short, X and Y referred to the X- and Y-coordinate, Z to elevation and g/d to the perceived quality of the sample (g-Good, d- doubtful, b- bad). We renamed the latter to "quality", see Table 2*

Lab_analysis_description.pdf: harmonized the language to English

*We provided a translation for the German descriptions and submitted the changes to the repository.*

*References:*

Dataflow Systems Ltd: Odyssey Capacitance Water Level Logger, [online] Available from: http://odysseydatarecording.com/index.php?route=product/product&path=59&product_id=50, 2021.

Kiewiet, L., Freyberg, J. and Meerveld, H. J. (Ilja): Spatiotemporal variability in hydrochemistry of shallow groundwater in a small pre-alpine catchment: The importance of landscape elements, Hydrol. Process., 33(19), 2502–2522, doi:10.1002/hyp.13517, 2019.

Kolbe, T., Marçais, J., de Dreuzy, J. R., Labasque, T. and Bishop, K.: Lagged rejuvenation of groundwater indicates internal flow structures and hydrological connectivity, Hydrol. Process., 34(10), 2176–2189, doi:10.1002/hyp.13753, 2020.

Laudon, H., Taberman, I., Ågren, A., Futter, M., Ottosson-Löfvenius, M. and Bishop, K.: The Krycklan Catchment Study - A flagship infrastructure for hydrology, biogeochemistry, and climate research in the boreal landscape, Water Resour. Res., 49(10), 7154–7158, doi:10.1002/wrcr.20520, 2013.

Laudon, H., Hasselquist, E. M., Peichl, M., Lindgren, K., Sponseller, R., Lidman, F., Kuglerová, L., Hasselquist, N. J., Bishop, K., Nilsson, M. B. and Ågren, A. M.: Northern landscapes in transition: Evidence, approach and ways forward using the Krycklan Catchment Study, Hydrol. Process., 35(4), 1–15, doi:10.1002/hyp.14170, 2021.

Penna, D. and van Meerveld, H. J.: Spatial variability in the isotopic composition of water in small catchments and its effect on hydrograph separation, Wiley Interdiscip. Rev. Water, 6(5), 1–33, doi:10.1002/wat2.1367, 2019.

Schwab, M., Klaus, J., Pfister, L. and Weiler, M.: Diel discharge cycles explained through viscosity fluctuations in riparian inflow, Water Resour. Res., (52), 8744–8755, doi:10.1002/2016WR018626, 2016.

**Reviewer II**

Summary and general comments:

In this manuscript, Erdbrügger et al. installed a monitoring network of groundwater wells in the Krycklan catchment in Northern Sweden and produced two datasets which combined hydrometric and hydrochemical data: groundwater levels and groundwater chemistry. The datasets can be a useful reference to test models that simulate groundwater dynamics and to evaluate subsurface hydrological connectivity. The manuscript is generally well-written and addresses two important datasets that is suitable for the Earth System Science Data. Thus, I can recommend a acceptance after a minor revision.

*We thank the reviewer for these positive words and the useful comments to which we respond in red font below.*

Line-to-line comments:

Page 1, line 9-10, Abstract, add a sentence to summarize the shortcomings of the existing research.

*We added such a sentence in the revised version of the manuscript (Lines 13-14)*

Page 1, line 10-21, Abstract, the methodology could be streamlined appropriately.

*We revised the abstract and streamlined the information.*

Tables, in general: tables in the manuscript should be spaced appropriately to fit the content; It is recommended to use three-line tables.

*We reformatted the tables to make the text more readable.*

Page 4, Table1, references can be placed in the last column.

*We moved the references to the last column.*

Page 4 and 5, Table1 and Table 2 can be used as an appendix.

*We moved Table 1 and 2 to the Appendix.*

Page 5, line 107, label the location of the Umea in Figure 1 or delete 'about 60 km inland from Umea'.

*We added the approximate coordinates of the Krycklan catchment (64°140N, 19°460E) and kept Umeå in the text as a local reference (Line 115). The position of Umeå on the map was too close to the catchment position, making the information unreadable. Therefore, we do not show the location of Umeå on the map of Sweden.*

Page 7, Figure 1 is missing the graticule.

*We omitted the graticule in Figure 1 to avoid clutter and information overload in the figure. We added the coordinates of the Krycklan catchment and provide the information on the map coordinate system. As all groundwater well coordinates are provided in the dataset, and the Krycklan catchment data are*

*openly available for download and further detailed analysis, we prefer to not include this information in this figure to keep it easier to "read".*

Figures in this manuscript (Figures 1, Figure 2, Figure 10, and Figure11), in general: reduce the size of the scale bar and north arrow to make the Figures more coordinated.

*We adjusted the sizes of the scale bar and North arrow in the figures.*

Page 17 and 19, Figure 6 and Figure 7, the labels in the Figures are too small to read.

*We increased the font size of the labels in both figures.*

---

## Author Response (AR2)

Revision 2

Reviewer's comments:

- Line 150: "with the networks of groundwater wells". This netwrok is not presented in fig.1

We meant to say that these are the areas where the networks are located but we see how it can be misleading. We removed the half sentence and rewrote the caption.

See line 150

- Lines 185-189: different range values between all wells (70 and 581) and the well in summer (57) and in winter (578). Should not be similar?

We thank the reviewer for spotting the error and fixed it. The minimum well depth is 57 cm and the maximum 581 cm.

See line 186-189

- Lines 466-467: with only one sampling during quite similar hydrological conditions not sure you can GW chemistry with GW level dynamics. I would remove this sentence. Or it was not clear to me.

We agree that this sentence is not clear as to its meaning there and removed it.

- Appendix B and online chemical data: the symbol for fluoride should be harmonized between the text (F in lines 422 and 449) and these two doucments (Flu)

We harmonized the text and adjusted the use of the symbol to "F" in the appendix and additional online data.